# Robust Recourse via Kernel Distributionally Robust Optimization and Bayesian Posterior Predictive Modeling

**Sita Bissu**                                                          *maz238455@maths.iitd.ac.in*
*Department of Mathematics*
*Indian Institute of Technology Delhi*

**Navnit Kumar Yadav**                                              *navnitydv@gmail.com*
*Department of Mathematics*
*Indian Institute of Technology Delhi*

**Aparna Mehra**                                                      *apmehra@maths.iitd.ac.in*
*Department of Mathematics*
*Indian Institute of Technology Delhi*

**Sandeep Kumar**                                                    *ksandeep@iitd.ac.in*
*Department of Electrical Engineering*
*Indian Institute of Technology Delhi*

**Reviewed on OpenReview:** *https://openreview.net/forum?id=LmEDkCTYOX*

## Abstract

Machine learning recourse provides actionable recommendations to achieve favorable outcomes from predictive decision models. A critical limitation of current approaches is their reliance on the assumption of model stationarity, an assumption that is frequently violated in dynamic, real-world settings with distributional shifts. Robust approaches such as Robust Algorithmic Recourse (ROAR) and the Wasserstein-based DiRRAc address some uncertainties but remain limited in handling nonlinear dependencies and large-scale shifts, including concept drift and the worst-case distributional shifts within the MMD ambiguity set. We propose Kernel Distributionally Robust Recourse Action (KDRRA), a framework that defines ambiguity sets using Maximum Mean Discrepancy (MMD) in a Reproducing Kernel Hilbert Space (RKHS), enabling flexible, nonparametric modeling of complex, nonlinear discrepancies between distributions. A practical challenge for kernel DRO is that empirical kernel mean embeddings can deviate from the true distribution, inflating ambiguity radii and yielding overly conservative recommendations. To address this, we introduce Bayesian KDRRA (BKDRRA), which centers the ambiguity set on a Bayesian posterior predictive distribution constructed via posterior bootstrap. This Bayesian centering integrates sampling variability and moderate model uncertainty into the reference distribution, leading to tighter ambiguity sets and markedly lower conservatism without sacrificing robustness. Leveraging the representer theorem, we derive finite-dimensional convex reformulations of the worst-case recourse optimization for both KDRRA and BKDRRA. We conduct a comprehensive empirical evaluation across four real-world datasets that exhibit correction, temporal, geospatial, and demographic covariate shifts. The KDRRA consistently outperforms state-of-the-art baselines in yielding superior robustness and lower recourse cost, while BKDRRA further improves stability and calibration by integrating Bayesian uncertainty. Our research advances the frontier of distributionally robust recourse by integrating machine learning tools and optimization, offering reliable and resilient decision-making under uncertainty.

**Keywords:** Actionable Recourse, Distributionally Robust Optimization, Reproducing Kernel Hilbert Space, Maximum Mean Discrepancy, Kernel Distributionally Robust Recourse Action, Bayesian Kernel Distributionally Robust Recourse Action

## 1  Introduction

Post-hoc explanations are essential in making machine learning systems transparent and trustworthy, especially when they are deployed in high-stakes domains such as credit scoring, healthcare triage, admissions, and resource allocation. Among these, *actionable recourse* has emerged as a particularly powerful form of local explanation: it prescribes concrete, feasible changes that an individual may implement to obtain a more favorable decision in the future. For instance, when a loan application is rejected, a recourse suggestion such as "increase monthly income by \$500" or "reduce current debt by 20%" informs the applicant of tangible steps to reverse the decision. Such forward-looking guidance strengthens user agency, supports human-AI collaboration, and helps preserve user trust in automated decision processes.

Despite its promise, generating reliable recourse remains challenging. A high-quality recourse action must satisfy three core criteria: *feasibility*, meaning the action should succeed in changing the model's prediction; *actionability*, meaning only mutable features may be altered; and *minimality*, meaning the required change should incur the smallest possible cost. Existing approaches address these goals through integer programming Ustun et al. (2019), gradient-based counterfactual search Karimi et al. (2021), multi-objective formulations Dandl et al. (2020), and diversity-promoting mechanisms Mothilal et al. (2020); Russell (2019). However, a critical assumption underlies nearly all of these methods: the predictive model is assumed to remain fixed after deployment.

This assumption is routinely violated in practice. Real-world data distributions evolve due to temporal, demographic, and environmental changes, prompting organizations to retrain or update deployed models. Such updates induce model-parameter shifts that can invalidate previously recommended recourse actions Rawal & Lakkaraju (2020). As a result, individuals who follow an earlier recourse recommendation may still receive an unfavorable outcome after the model changes, undermining trust in automated decision systems Rudin (2019); Venkatasubramanian & Alfano (2020).

Recent work has begun to address this challenge through robust recourse formulations. ROAR introduces robustness via norm-bounded uncertainty sets over model parameters Upadhyay et al. (2021), while DiRRAc proposes a distributionally robust formulation using Wasserstein ambiguity sets over classifier parameters Nguyen et al. (2023). Although these approaches represent important progress, they suffer from two fundamental limitations. First, they are largely restricted to linear or structured parametric models, limiting their applicability to modern nonlinear decision systems. Second, their uncertainty sets rely on rigid geometries such as boxes or ellipsoids that struggle to capture complex, distribution-level shifts induced by retraining, subsampling, or heterogeneous data sources.

To overcome these limitations, we propose KDRRA, a new framework that models uncertainty using Maximum Mean Discrepancy (MMD) in a RKHS  Gretton et al. (2012); Berlinet & Thomas-Agnan (2011). Rather than constraining uncertainty directly in the parameter space, KDRRA embeds the distribution of future classifier parameters into an RKHS and constructs an MMD-based ambiguity set around its kernel mean embedding Muandet et al. (2017). This kernelized representation captures rich nonlinear discrepancies between distributions while remaining amenable to convex optimization through representer-theorem-based reductions Staib & Jegelka (2019); Zhu et al. (2021). As a result, KDRRA enables principled robustness to complex, nonlinear distributional shifts that invalidate existing recourse methods.

A key challenge in kernel-based distributionally robust optimization is the reliance on empirical kernel mean embeddings. When data are limited or subject to sampling bias, these embeddings can deviate substantially from the true underlying distribution, forcing large ambiguity radii and resulting in overly conservative solutions Staib & Jegelka (2019). To address this issue, we introduce BKDRRA. Instead of centering the ambiguity set at the empirical distribution, BKDRRA centers it at a Bayesian posterior predictive distribution constructed via the Bayesian bootstrap and posterior predictive averaging Newton & Raftery (1994); Rubin (1981); Lyddon et al. (2018). This predictive distribution integrates sampling variability and

moderate model misspecification directly into the ambiguity-set center, allowing for tighter uncertainty sets and substantially reduced conservatism while preserving robustness guarantees.

Together, KDRRA and BKDRRA provide a unified framework for distribution-aware recourse under uncertain future model behavior. KDRRA represents a fully data-driven, nonparametric robustness approach, while BKDRRA incorporates probabilistic uncertainty through Bayesian averaging. We evaluate both methods on four real-world datasets exhibiting temporal, correction, geospatial, and demographic covariate shifts. Our results demonstrate that KDRRA achieves strong robustness under nonlinear distributional changes and that BKDRRA further improves efficiency by reducing recourse cost without sacrificing reliability.

**Research Contributions.** This work advances reliable, distribution-aware recourse by integrating kernel methods, distributional robustness, and Bayesian uncertainty modeling. Our main contributions are as follows:

1. We introduce KDRRA, a novel recourse framework that models nonlinear distributional shifts through MMD in a Reproducing Kernel Hilbert Space (RKHS). Instead of constraining uncertainty in the parameter space, KDRRA constructs an RKHS-based ambiguity set around the kernel mean embedding of the future model distribution, enabling principled robustness to complex nonlinear shifts that arise in modern machine-learning models.

2. We derive an explicit finite-dimensional formulation for KDRRA by expressing the MMD ambiguity constraint through Gram matrix representations of RKHS norms and upper-bounding the violation indicators, which yields a tractable quadratic constraint program (QCP) that is efficiently solvable using standard solvers, thereby providing the first tractable MMD-based recourse formulation.

3. To mitigate the conservatism inherent in empirical kernel ambiguity sets, we propose a Bayesian extension that centers the MMD ball at the *Bayesian posterior predictive distribution*. By leveraging the posterior bootstrap or nonparametric learning, BKDRRA internalizes both sampling variability and moderate model misspecification into the ambiguity center, producing significantly tighter and more reliable recourse actions.

4. We show that KDRRA and BKDRRA constitute two complementary regimes of uncertainty modeling, one fully data-driven and one probabilistically informed, providing a unified framework that interpolates between agnostic distributional robustness and Bayesian decision-making.

5. Through extensive empirical experiments under nonlinear, temporal, and geospatial shifts on four real-world benchmarks exhibiting realistic distribution shifts, we demonstrate that KDRRA substantially improves robustness under nonlinear perturbations, while BKDRRA further reduces conservatism and enhances recourse validity. Comparisons against AR, Wachter counterfactuals, ROAR, DiRRAc, and Gaussian-DiRRAc confirm the superiority of our proposed methods.

The rest of the paper is organized as follows. Section 2 reviews the literature on algorithmic recourse, robust and distributionally robust formulations, kernel-based distributional robustness, and Bayesian uncertainty modeling. Section 3 formalizes the recourse-action problem under model shifts and introduces the corresponding distributionally robust formulation. Section 4 presents the proposed Kernel Distributionally Robust Recourse Action (KDRRA) framework, including the construction of RKHS-based ambiguity sets using Maximum Mean Discrepancy, tractable reformulations of the worst-case violation constraints, and finite-sample guarantees. Section 5 introduces the Bayesian KDRRA (BKDRRA) framework, in which the ambiguity-set center is replaced by a Bayesian posterior predictive distribution obtained via posterior bootstrap. The section further develops finite-dimensional dual representations and representer-theorem-based reductions, culminating in second-order cone programming formulations, along with finite-sample coverage guarantees. Section 6 reports experimental results on four real-world datasets: German Credit, SBA Loans, Student Performance, and Adult Income, and additionally evaluates performance on nonlinear classifiers using local linear approximations via LIME, demonstrating improved robustness and reduced recourse cost compared to existing baselines. Section 7 concludes the paper and discusses future research directions, with additional sensitivity analysis, runtime comparisons, proofs, and algorithmic details provided in the appendix.

# 2 Literature Overview

We organize the literature on algorithmic recourse and robustness according to how uncertainty in predictive models is represented and addressed. The first line of work studies recourse under fixed predictive models, with an emphasis on feasibility, actionability, and minimality of prescribed actions. A second and more recent body of work investigates robust and distributionally robust recourse, aiming to preserve recourse validity under model updates or distributional shifts. Third, kernel-based approaches to distributional robustness construct nonparametric ambiguity sets using reproducing kernel Hilbert spaces and statistical discrepancies such as maximum mean discrepancy. Finally, Bayesian approaches model uncertainty through posterior or posterior predictive distributions, offering probabilistic alternatives to purely worst-case robustness. We review each of these strands in turn and clarify how their respective assumptions and limitations motivate our proposed framework.

## 2.1 Algorithmic Recourse under Fixed Predictive Models

Algorithmic recourse aims to provide individuals with actionable changes to their features that would lead to a favorable prediction from a deployed model. Early foundational work formalized recourse for linear classifiers using integer programming and cost-minimization objectives, explicitly accounting for feature mutability and actionability constraints Ustun et al. (2019). Subsequent approaches extended recourse generation beyond linear settings through gradient-based counterfactual search Karimi et al. (2021), multi-objective optimization balancing feasibility and cost Dandl et al. (2020), and diversity-promoting mechanisms that return multiple alternative actions Russell (2019); Mothilal et al. (2020). Recent work has further emphasized that claims of actionability in recourse methods are often underspecified, proposing human-centered evaluation tools to systematically assess how actionable recourse information is perceived by users across contexts Singh et al. (2025).

Despite methodological differences, a unifying assumption across these approaches is that the predictive model remains fixed after recourse is issued. This assumption significantly limits their reliability in real-world deployments, where retraining, data drift, and policy updates are common. Several works have emphasized the practical and philosophical implications of this limitation, noting that recourse recommendations may lose validity once the underlying decision rule changes Rawal & Lakkaraju (2020); Rudin (2019); Venkatasubramanian & Alfano (2020). Our work directly addresses this failure mode by explicitly modeling uncertainty in future classifier behavior.

## 2.2 Robust and Distributionally Robust Recourse

To mitigate the fragility of recourse under model updates, recent studies have introduced robustness into the recourse generation process. ROAR proposes a robust optimization framework that enforces recourse validity under norm-bounded uncertainty sets over classifier parameters Upadhyay et al. (2021). While effective for linear models, ROAR relies on box- or ellipsoid-shaped uncertainty sets and does not capture distributional or nonlinear variations arising from retraining or heterogeneous data sources.

More recently, DiRRAc formulates recourse as a distributionally robust optimization problem by constructing Wasserstein ambiguity sets over classifier parameters Nguyen et al. (2023). This approach allows uncertainty to be modeled at the distribution level and admits tractable reformulations under linear or Gaussian assumptions. However, Wasserstein-based ambiguity sets impose rigid transport geometries that limit their ability to represent complex, nonlinear discrepancies between parameter distributions. Moreover, tractability is achieved only under restrictive structural assumptions, limiting applicability to modern, high-dimensional settings.

In contrast to these approaches, our work constructs ambiguity sets using MMD in an RKHS, enabling nonparametric and nonlinear modeling of distributional shifts while preserving tractability through kernel-based dual representations.

### 2.3 Kernel-Based Distributionally Robust Optimization

Kernel methods provide a powerful framework for representing probability distributions via kernel mean embeddings in RKHS Gretton et al. (2012); Berlinet & Thomas-Agnan (2011). MMD has been widely used as a statistical distance for two-sample testing and distribution comparison Gretton et al. (2012). Building on these foundations, kernel-based distributionally robust optimization has emerged as a flexible alternative to Wasserstein DRO Mohajerin Esfahani & Kuhn (2018); Kuhn et al. (2019), allowing ambiguity sets to capture nonlinear distributional discrepancies Smola et al. (2006); Borgwardt et al. (2006).

Existing kernel DRO methods primarily focus on prediction, generalization, or risk minimization Staib & Jegelka (2019); Liu et al. (2021b;a), and do not address recourse generation, where decision variables correspond to actionable feature changes rather than model parameters or predictions. Furthermore, prior work typically centers kernel ambiguity sets at empirical kernel mean embeddings, which can be unstable under limited data or sampling bias. Our KDRRA framework adapts kernel DRO principles to algorithmic recourse while addressing empirical instability through Bayesian centering.

### 2.4 Bayesian Uncertainty and Posterior Predictive Robustness

Bayesian approaches provide a principled mechanism for modeling uncertainty in data-generating processes and learned models. The Bayesian bootstrap and weighted likelihood bootstrap offer nonparametric tools for approximating posterior uncertainty without explicit likelihood specification Newton & Raftery (1994); Rubin (1981); Lyddon et al. (2018). More recent works have shown that posterior predictive distributions can serve as robust reference models by integrating sampling variability and moderate model misspecification Huang et al. (2023); Liu & Briol (2025); Wehenkel et al. (2024).

In the context of distributionally robust optimization, Bayesian ideas have been explored to hedge against model uncertainty or to approximate ambiguity sets through parametric assumptions $\widetilde{P}$ Iyengar et al. (2023); Michel et al. (2021). However, these methods are not designed for recourse generation and do not leverage kernel-based representations to capture nonlinear distributional shifts. To the best of our knowledge, no prior work integrates Bayesian posterior predictive modeling with kernel-based DRO for algorithmic recourse.

### 2.5 Synthesis and Research Gap

Existing recourse methods either ignore uncertainty in future model behavior or rely on rigid uncertainty sets that fail to capture nonlinear distributional shifts. Robust and distributionally robust recourse approaches improve reliability under restricted forms of uncertainty but struggle to capture nonlinear, distribution-level shifts induced by realistic retraining procedures. Kernel-based DRO provides expressive, nonparametric ambiguity sets but has not been applied to recourse problems nor combined with Bayesian posterior predictive modeling to control conservatism.

Our work bridges these gaps by introducing KDRRA, which leverages RKHS-based MMD ambiguity sets to model complex distributional shifts in future classifiers, and BKDRRA, which further reduces conservatism by centering ambiguity sets at a Bayesian posterior predictive distribution. Together, these methods provide a unified framework for distribution-aware, nonparametric, and uncertainty-calibrated algorithmic recourse.

## 3 Problem Definition

Consider a binary classification problem where the label 0 denotes an unfavorable outcome for the user, and label 1 denotes a favorable outcome. The input space is $R^n$. A linear classifier $C_\theta : \mathbb{R}^n \to \{0, 1\}$, parameterized by $\theta \in \mathbb{R}^n$, assigns a label to any input $x \in \mathbb{R}^n$ based on the sign of the linear score $\theta^\top x$,

$$C_\theta(x) = \mathbb{I}(\theta^\top x \geq 0) = \begin{cases} 1, & \text{if } \theta^\top x \geq 0, \\ 0, & \text{if } \theta^\top x < 0, \end{cases}$$

where $\mathbb{I}(\cdot)$ denotes the indicator function.

Note that an intercept term can be incorporated by adding an extra dimension in $\theta$.

At the current time $t = 0$, the classifier is parameterized by a known vector $\theta_0 \in \mathbb{R}^n$, and the user's present action $x_0$ leads to an unfavorable prediction.

$$C_{\theta_0}(x_0) = 0.$$

At a future time $t = 1$, the classifier may change due to model updates or distributional drift. We denote its future parameter by the random vector $\tilde{\theta} \in \mathbb{R}^n$. The user now seeks a future action $x$ that remains close to the current action $x_0$ according to a prescribed distance measure and is classified favorably with high probability under the uncertain future classifier.

This provides the following recourse optimization problem:

$$
\begin{aligned}
\min_{x \in \mathcal{X}} \quad & d(x, x_0) \\
\text{s.t.} \quad & \mathbb{P}(C_{\tilde{\theta}_k}(x) > 0) \geq 1 - \delta, \quad \forall\, k \in [K],
\end{aligned}
\tag{1}
$$

Here, $d(x, x_0)$ denotes a chosen distance measure on the action space $\mathcal{X}$; $[K]$ represents the index set of possible parameter shifts of the classifier; $\mathcal{X}$ is the feasible (admissible) action space; $\delta > 0$ is a prescribed tolerance parameter; and $\tilde{\theta}_k$ denotes the classifier parameter at time $t = 1$ under shift $k$.

The future parameter $\tilde{\theta}_k$ is random as it is derived from retraining the classifier on a distributionally shifted dataset. Because each realization of this shifted training data generates a unique parameter estimate, the resulting ensemble of these estimates defines the probability distribution for $\tilde{\theta}_k$.

## 3.1 Distributionally Robust Recourse Action

Since the exact distribution of $\tilde{\theta}_k$ cannot be predetermined, we adopt a distributionally robust formulation. For each shift $k \in [K]$, we introduce an ambiguity set $\mathcal{P}_k$, containing all plausible distributions over $\tilde{\theta}_k$, and require the recourse action to be feasible under every distribution in $\mathcal{P}_k$. Formally, the distributionally robust recourse problem is:

$$
\begin{aligned}
\min_{x \in \mathcal{X}} \quad & d(x, x_0), \\
\text{s.t.} \quad & \inf_{P \in \mathcal{P}_k} \mathbb{P}_{\tilde{\theta}_k \sim P}\big(\tilde{\theta}_k^\top x \geq 0\big) \geq 1 - \delta, \qquad \forall\, k \in [K].
\end{aligned}
$$

The constraint requires that, under the worst-case distribution in the ambiguity set $\mathcal{P}_k$, the probability that the linear score $\tilde{\theta}_k^\top x$ is non-negative is at least $1 - \delta$.

Equivalently, by bounding the worst-case probability of an unfavorable outcome:

$$
\begin{aligned}
\min_{x \in \mathcal{X}} \quad & d(x, x_0), \\
\text{s.t.} \quad & \sup_{P \in \mathcal{P}_k} \mathbb{P}_{\tilde{\theta}_k \sim P}\big(\tilde{\theta}_k^\top x < 0\big) \leq \delta, \qquad \forall\, k \in [K].
\end{aligned}
\tag{2}
$$

The constraint in problem 2 requires that even under the worst-case distribution in $\mathcal{P}_k$, the probability of an unfavorable classification remains at most $\delta$. The key modeling choice is therefore the construction of $\mathcal{P}_k$, which we now develop using kernel mean embeddings and the Maximum Mean Discrepancy (MMD).

## 3.2 Uncertainty Set Construction via RKHS

To model distributional uncertainty in problem 2, we construct ambiguity sets for the shifted classifier parameters $\tilde{\theta}_k$. Since each $\tilde{\theta}_k$ takes values in a common parameter space $\Theta$, we adopt a kernel-based representation of probability distributions on $\Theta$ using RKHS embeddings. This framework enables a flexible and nonparametric characterization of distributional shifts via the MMD.

**Definition 3.1** (Reproducing Kernel Hilbert Space)**.** Let $\Theta$ be a nonempty set. A function $\kappa : \Theta \times \Theta \to \mathbb{R}$ is called a *positive definite kernel* if

1. it is symmetric, i. e., $\kappa(\theta_i, \theta_j) = \kappa(\theta_j, \theta_i)$ for all $\theta_i, \theta_j \in \Theta$, and

2. for every $n \in \mathbb{N}$, every choice of points $\theta_1, \ldots, \theta_n \in \Theta$, and every set of coefficients $c_1, \ldots, c_n \in \mathbb{R}$, the Gram matrix is positive semidefinite:

$$\sum_{i=1}^{n} \sum_{j=1}^{n} c_i c_j \, \kappa(\theta_i, \theta_j) \geq 0.$$

A positive definite kernel induces a Hilbert space of real-valued functions on $\Theta$, called RKHS, can be denoted by $\mathcal{H}$.

By the Moore-Aronszajn theorem Cicekyurt (2025), for any positive definite kernel $\kappa$ there exists a unique RKHS $\mathcal{H}$ and a feature map $\phi : \Theta \to \mathcal{H}$ such that,

$$f(\theta) = \langle f, \, \kappa(\theta, \cdot) \rangle_{\mathcal{H}} = \langle f, \, \phi(\theta) \rangle_{\mathcal{H}}, \qquad \forall \, f \in \mathcal{H}, \, \theta \in \Theta.$$

This is known as the *reproducing property.* Moreover, the kernel admits the representation

$$\kappa(\theta_i, \theta_j) = \langle \phi(\theta_i), \, \phi(\theta_j) \rangle_{\mathcal{H}} = \langle \kappa(\theta_i, \cdot), \, \kappa(\theta_j, \cdot) \rangle_{\mathcal{H}}, \qquad \forall \, \theta_i, \, \theta_j \in \Theta.$$

**Definition 3.2** (Kernel Mean Embedding)**.** Muandet et al. (2017) The *kernel mean embedding* (KME) is a non-parametric representation of probability distributions in an RKHS. For a probability measure $\mathbb{P}$ on $\Theta$ and a kernel $\kappa$, the KME of $P$ is defined as

$$\mu_{\mathbb{P}} := \int \kappa(\theta, \cdot) \, d\mathbb{P}(\theta).$$

Intuitively, the KME represents the expectation of the feature map under $\mathbb{P}$ i. e.,

$$\mu_{\mathbb{P}} = \mathbb{E}_{\Theta \sim \mathbb{P}}[\phi(\Theta)].$$

By standard results in kernel mean embedding theory (Muandet et al. (2017)), the embedding $\mu_{\mathbb{P}}$ exists for any $\mathbb{P}$ satisfying $\int \sqrt{\kappa(\theta, \theta)} \, d\mathbb{P}(\theta) < \infty$.

**Definition 3.3** (Maximum Mean Discrepancy)**.** (Smola et al. (2006); Borgwardt et al. (2006))

Given two probability measures $\mathbb{P}$ and $\mathbb{Q}$ on $\Theta$ with respective kernel mean embeddings $\mu_{\mathbb{P}}, \, \mu_{\mathbb{Q}} \in \mathcal{H}$, the MMD is defined as

$$\mathrm{MMD}(\mathbb{P}, \mathbb{Q}) := \|\mu_{\mathbb{P}} - \mu_{\mathbb{Q}}\|_{\mathcal{H}}.$$

Using the reproducing property, the squared MMD can be expressed in terms of the kernel $\kappa$:

$$
\begin{aligned}
\|\mu_{\mathbb{P}} - \mu_{\mathbb{Q}}\|_{\mathcal{H}}^2 &= \langle \mu_{\mathbb{P}} - \mu_{\mathbb{Q}}, \, \mu_{\mathbb{P}} - \mu_{\mathbb{Q}} \rangle_{\mathcal{H}} \\
&= \langle \mu_{\mathbb{P}}, \mu_{\mathbb{P}} \rangle_{\mathcal{H}} + \langle \mu_{\mathbb{Q}}, \mu_{\mathbb{Q}} \rangle_{\mathcal{H}} - 2 \langle \mu_{\mathbb{P}}, \mu_{\mathbb{Q}} \rangle_{\mathcal{H}} \\
&= \mathbb{E}_{\theta_i, \theta_i' \sim \mathbb{P}}[\kappa(\theta_i, \theta_i')] + \mathbb{E}_{\theta_j, \theta_j' \sim \mathbb{Q}}[\kappa(\theta_j, \theta_j')] - 2\mathbb{E}_{\theta_i \sim \mathbb{P}, \, \theta_j \sim \mathbb{Q}}[\kappa(\theta_i, \theta_j)].
\end{aligned}
\tag{3}
$$

The expectations above can be estimated empirically using samples from $\mathbb{P}$ and $\mathbb{Q}$ (Gretton et al. (2012)).

The RKHS embedding framework above provides a principled way to quantify discrepancies between probability distributions on $\Theta$. We now use it to construct the ambiguity sets $\mathcal{P}_k$ for problem 2. Following Zhu et al. (2021) and Staib & Jegelka (2019), we define a generic MMD ball centered at a reference distribution $\mathbb{P} \in \mathcal{P}(\Theta)$ with radius $\varepsilon > 0$ as:

$$\mathcal{K}_{\varepsilon}(\mathbb{P}) = \{\mathbb{Q} \in \mathcal{P}(\Theta) \, : \, \|\mu_{\mathbb{Q}} - \mu_{\mathbb{P}}\|_{\mathcal{H}} \leq \varepsilon\}. \tag{4}$$

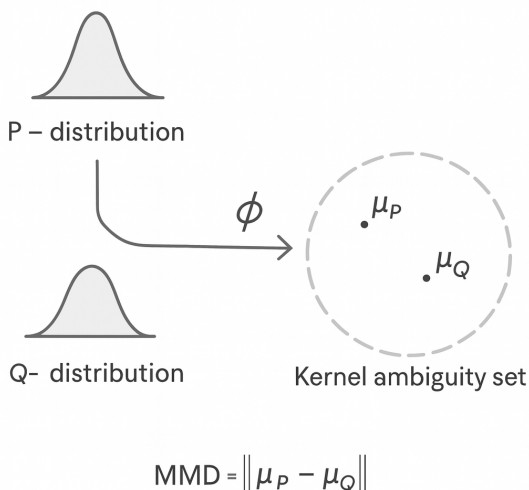

Figure 1: Mean embeddings $\mu_{\mathbb{P}}$ and $\mu_{\mathbb{Q}}$ of distributions $\mathbb{P}$ and $\mathbb{Q}$ under the feature map $\phi$.

We now specialize this construction to each uncertain parameter $\tilde{\theta}_k$ in problem 2. For each shift $k \in [K]$, let $P_k$ denote the reference distribution over $\tilde{\theta}_k$, estimated from observed parameter realizations under shift $k$. The ambiguity set for shift $k$ is defined as:

$$\mathcal{P}_k := \mathcal{K}_{\varepsilon_k}(\mathbb{P}_k) = \left\{ \mathbb{Q} \in \mathcal{P}(\Theta) : \|\mu_{\mathbb{Q}} - \mu_{\mathbb{P}_k}\|_{\mathcal{H}} \leq \varepsilon_k \right\}, \tag{5}$$

where $\mu_{\mathbb{P}_k} = \int \kappa(\theta, \cdot)\, d\mathbb{P}_k(\theta)$ is the KME of the reference distribution $\mathbb{P}_k$, and $\varepsilon_k > 0$ is the user-specified ambiguity radius for shift $k$. Thus $\mathcal{P}_k$ contains all probability distributions over future classifier parameters whose RKHS embeddings lie within an $\varepsilon_k$-ball around the reference embedding $\mu_{\mathbb{P}_k}$.

Substituting $\mathcal{P}_k$ from 5 into the worst-case constraint of problem 2 and reformulating it tractably using the RKHS structure, together with finite-sample guarantees, is the subject of the next section.

## 4 Kernel Distributionally Robust Recourse Action

With the per-shift ambiguity set $\mathcal{P}_k$ defined in 5, the worst-case constraint in problem 2 for a fixed shift $k \in [K]$ and action $x \in \mathcal{X}$ becomes:

$$\sup_{\mathbb{P} \in \mathcal{P}_k} \mathbb{P}_{\tilde{\theta}_k \sim P}\left( \tilde{\theta}_k^\top x < 0 \right) \leq \delta.$$

To reformulate this tractably, we represent the distribution $\mathbb{P}$ through its KME. Specifically, for any $\mathbb{P} \in \mathcal{P}(\Theta)$, its KME is $\mu = \int \phi(u)\, d\mathbb{P}(u) \in \mathcal{H}$, and membership of $P$ in $\mathcal{P}_k = \mathcal{K}_{\varepsilon_k}(\mathbb{P}_k)$ is equivalent to requiring $\|\mu - \mu_{\mathbb{P}_k}\|_{\mathcal{H}} \leq \varepsilon_k$. The worst-case violation problem for shift $k$ therefore admits the following equivalent RKHS representation:

$$\sup_{\mathbb{P}} \ \mathbb{P}\left( C_{\tilde{\theta}_k}(x) \leq 0 \right) \quad \text{s.t.} \quad \int \phi(u)\, d\mathbb{P}(u) = \mu, \quad \|\mu - \mu_{\mathbb{P}_k}\|_{\mathcal{H}} \leq \varepsilon_k, \tag{6}$$

The first constraint encodes that $\mu$ is the mean embedding of $P$; the second enforces that $\mathbb{P}$ lies within the MMD ball of radius $\varepsilon_k$ around the reference $\mathbb{P}_k$. The reference embedding $\mu_{\mathbb{P}_k}$ is estimated empirically from observed parameter realizations $\{\beta_i\}_{i=1}^N$ under shift $k$ as:

$$\hat{\mu}_{\mathbb{P}_k} = \frac{1}{N} \sum_{i=1}^N \phi(\beta_i).$$

To obtain a computationally tractable formulation of problem 6, we approximate both $\mu$ and $\mu_{\mathbb{P}_k}$ using finite empirical support. Let $\{\beta_i\}_{i=1}^N$ be the collected samples from the distribution of $\tilde{\theta}_k$ under shift $k$, and let $\{\gamma_j\}_{j=1}^Z$ be additional synthetic samples generated near the decision boundary $\{\theta : \theta^\top x = 0\}$ via perturbation or convex combination. Define the combined atom set:

$$\{\alpha_i\}_{i=1}^M = \{\beta_1, \ldots, \beta_N, \gamma_1, \ldots, \gamma_Z\}, \qquad M = N + Z.$$

**Remark:** [Role of real and synthetic samples] The observed samples $\{\beta_i\}_{i=1}^N$ anchor the reference embedding $\hat{\mu}_{\mathbb{P}_k}$ and define the center of the MMD ball. The synthetic samples $\{\gamma_j\}_{j=1}^Z$ are necessary because the worst-case probability distribution $P$ in problem 6 may place mass on parameter values not present in $\{\beta_i\}$ yet lying within the MMD ball, in particular, on parameters near the decision boundary $\{\theta : \theta^\top x = 0\}$ where the recourse action $x$ is most vulnerable to failure. Restricting $\mathbb{P}$ to the support of $\{\beta_i\}$ alone would systematically underestimate the true worst-case violation probability.

Under this discrete approximation, the worst-case probability distribution is parameterized by a weight vector $\mathbf{c} = [c_1, \ldots, c_M]^\top$ with $c_i \geq 0$ and $\sum_{i=1}^M c_i = 1$, yielding:

$$\mu = \sum_{i=1}^M c_i\, \phi(\alpha_i), \qquad \hat{\mu}_{\mathbb{P}_k} = \frac{1}{N} \sum_{i=1}^N \phi(\beta_i).$$

With this parameterization, the RKHS norm constraint in problem 6 reduces to a finite-dimensional quadratic inequality in $\mathbf{c}$, as established in the following lemma.

**Lemma 4.1.** *The RKHS-norm constraint*

$$\left\| \sum_{i=1}^M c_i \phi(\alpha_i) - \frac{1}{N} \sum_{i=1}^N \phi(\beta_i) \right\|_{\mathcal{H}} \leq \epsilon$$

*is equivalent to the quadratic inequality*

$$\mathbf{c}^\top \mathcal{K}_\alpha \mathbf{c} - \frac{2}{N} \mathbf{c}^\top \mathcal{K}_{\alpha\beta} \mathbb{1} + \frac{1}{N^2} \mathbb{1}^\top \mathcal{K}_\beta \mathbb{1} \leq \epsilon^2, \tag{7}$$

*where $\mathbf{c} = [c_1, \ldots, c_M]^\top$, $\mathbb{1}$ is the all-ones vector, and $\mathcal{K}_\alpha, \mathcal{K}_{\alpha\beta}, \mathcal{K}_\beta$ are the appropriate Gram matrices.*

*Proof.* The proof is provided in Appendix C.2. $\qquad\square$

**Proposition 4.2.** *For a fixed action $x$ and shift $k$, the worst-case violation problem 6 over the kernel ambiguity set admits the finite-dimensional formulation*

$$\begin{aligned}
\max_{\mathbf{c} \in \mathbb{R}^M} \quad & \mathcal{I}(x)^\top \mathbf{c} \\
s.t. \quad & \mathbf{c}^\top \mathcal{K}_\alpha \mathbf{c} - \frac{2}{N} \mathbf{c}^\top \mathcal{K}_{\alpha\beta} \mathbb{1} + \frac{1}{N^2} \mathbb{1}^\top \mathcal{K}_\beta \mathbb{1} \leq \epsilon^2, \\
& \mathbb{1}^\top \mathbf{c} = 1, \\
& \mathbf{c} \geq 0,
\end{aligned} \tag{8}$$

*where*

$$\mathcal{I}(x) = \left[ \mathbb{I}(C_{\alpha_1}(x) \leq 0), \ldots, \mathbb{I}(C_{\alpha_M}(x) \leq 0) \right]^\top,$$

.

*Proof.* A detailed proof can be found in Appendix C.3. $\qquad\square$

**Lemma 4.3.** *Let $\mathcal{K}_\alpha \succeq 0$ (Positive semi definite) and let $R$ be a matrix such that $R^\top R = \mathcal{K}_\alpha$. Then the quadratic constraint*

$$\mathbf{c}^\top \mathcal{K}_\alpha \mathbf{c} - \frac{2}{N} \mathbf{c}^\top \mathcal{K}_{\alpha\beta} \mathbb{1} + \frac{1}{N^2} \mathbb{1}^\top \mathcal{K}_\beta \mathbb{1} \leq \epsilon^2$$

*in problem 8 is equivalent to the Lorentz cone constraint*

$$
\begin{bmatrix} R\mathbf{c} \\ \frac{1}{N}\mathbb{1}^\top \mathcal{K}_{\alpha\beta}\mathbf{c} \\ \frac{1}{N}\mathbb{1}^\top \mathcal{K}_{\alpha\beta}\mathbf{c} \end{bmatrix} - \begin{bmatrix} \mathbb{0} \\ \frac{1}{2N^2}\mathbb{1}^\top \mathcal{K}_\beta\mathbb{1} - \frac{\epsilon^2}{2} + \frac{1}{2} \\ \frac{1}{2N^2}\mathbb{1}^\top \mathcal{K}_\beta\mathbb{1} - \frac{\epsilon^2}{2} - \frac{1}{2} \end{bmatrix} \in \mathcal{Q}_{M+2},
$$

*where $\mathcal{Q}_{M+2}$ denotes the $(M+2)$-dimensional Lorentz cone. Consequently, the worst-case violation problem 8 admits an SOCP (Second Order Cone Program) representation.*

*Proof.* See Appendix C.4. □

**Proposition 4.4.** *Consider the above SOCP in Proposition 4.3, its conic dual can be written as*

$$
\min_{w,a_1,a_2,\rho,\lambda} \quad -\left(\frac{1}{2N^2}\mathbb{1}^\top \mathcal{K}_\beta\mathbb{1} - \frac{\epsilon^2}{2} + \frac{1}{2}\right)a_1 - \left(\frac{1}{2N^2}\mathbb{1}^\top \mathcal{K}_\beta\mathbb{1} - \frac{\epsilon^2}{2} - \frac{1}{2}\right)a_2 - \lambda,
$$
$$
\text{s.t.} \quad \frac{1}{N}\mathcal{K}_{\alpha\beta}\mathbb{1}\, a_1 + \frac{1}{N}\mathcal{K}_{\alpha\beta}\mathbb{1}\, a_2 + I^\top \rho + \mathbb{1}\lambda + R^\top w = -\mathcal{I}(x), \tag{9}
$$
$$
\|w\|^2 + a_1^2 \le a_2^2,
$$
$$
\rho_j \ge 0, \qquad j = 1,\ldots,M,
$$

*and, since $\rho \ge 0$, it admits the equivalent reduced form*

$$
\min_{w,a_1,a_2,\lambda} \quad -\left(\frac{1}{2N^2}\mathbb{1}^\top \mathcal{K}_\beta\mathbb{1} - \frac{\epsilon^2}{2} + \frac{1}{2}\right)a_1 - \left(\frac{1}{2N^2}\mathbb{1}^\top \mathcal{K}_\beta\mathbb{1} - \frac{\epsilon^2}{2} - \frac{1}{2}\right)a_2 - \lambda,
$$
$$
\text{s.t.} \quad \frac{1}{N}\mathcal{K}_{\alpha\beta}\mathbb{1}\, a_1 + \frac{1}{N}\mathcal{K}_{\alpha\beta}\mathbb{1}\, a_2 + \mathbb{1}\lambda + R^\top w \le -\mathcal{I}(x), \tag{10}
$$
$$
\|w\|^2 + a_1^2 \le a_2^2.
$$

*Proof.* The proof is detailed in Appendix C.5. □

**Theorem 4.5.** *Consider the worst-case constraint in 2 under the kernel ambiguity set 5. The kernel distributionally robust recourse problem 2 is equivalent to*

$$
\min_{x\in\mathcal{X},\,\{a_{1k},a_{2k},w_k,\lambda_k\}_{k=1}^K} \quad d(x,x_0)
$$
$$
\text{s.t.} \quad -\left(\frac{1}{2N_k^2}\mathbb{1}^\top \mathcal{K}_\beta^{(k)}\mathbb{1} - \frac{\epsilon_k^2}{2} + \frac{1}{2}\right)a_{1k} - \left(\frac{1}{2N_k^2}\mathbb{1}^\top \mathcal{K}_\beta^{(k)}\mathbb{1} - \frac{\epsilon_k^2}{2} - \frac{1}{2}\right)a_{2k} - \lambda_k \;\le\; \delta, \;\; \forall\, k \in [K],
$$
$$
\frac{1}{N_k}\mathcal{K}_{\alpha\beta}^{(k)}\mathbb{1}\, a_{1k} + \frac{1}{N_k}\mathcal{K}_{\alpha\beta}^{(k)}\mathbb{1}\, a_{2k} + \mathbb{1}\lambda_k + R_k^\top w_k \;\le\; -\mathcal{I}_k(x), \quad \forall\, k \in [K],
$$
$$
\|w_k\|^2 + a_{1k}^2 \le a_{2k}^2, \quad \forall\, k \in [K]. \tag{11}
$$

*where*

$$
\mathcal{I}_k(x) = \left[\mathbb{I}(C_{\alpha_1^{(k)}}(x) \le 0),\ldots,\mathbb{I}(C_{\alpha_{M_k}^{(k)}}(x) \le 0)\right]^\top.
$$

*The dependence of $\mathcal{I}_k(x)$ on $x$ induces binary decisions, and hence 11 is a mixed-integer quadratically constrained program (MIQCP).*

*Proof.* For a fixed shift $k$ and action $x$, the worst-case violation probability over the kernel ambiguity set admits the finite-dimensional formulation 8 via empirical support reduction. By Lemma 4.3, the resulting RKHS constraint is equivalently represented as a second-order cone constraint, enabling conic dualization. Applying SOCP duality (Proposition 4.4) and substituting the reduced dual form into the worst-case constraint of 2 for all $k \in [K]$ yields the finite-dimensional formulation 11. Since the violation vector $\mathcal{I}_k(x)$ depends on $x$ through indicator functions, optimizing jointly over $x$ and the dual variables induces binary decisions, resulting in an MIQCP. □

**Corollary 4.6.** Let $\psi : \mathbb{R} \to \mathbb{R}_+$ be convex and satisfy $\mathbb{I}(t \leq 0) \leq \psi(t)$ for all $t \in \mathbb{R}$. Define the surrogate indicator vector

$$\widetilde{\mathcal{I}}_k(x) = \left[ \psi\left( C_{\alpha_1^{(k)}}(x) \right), \ldots, \psi\left( C_{\alpha_{M_k}^{(k)}}(x) \right) \right]^\top.$$

Then, replacing $\mathcal{I}_k(x)$ by its surrogate $\widetilde{\mathcal{I}}_k(x)$ in Theorem 4.5 yields a tractable restriction of the KDRRA feasible set. In particular, when $C_{\alpha_i^{(k)}}(x)$ is affine in $x$ and $\mathcal{X}$ is convex, the resulting optimization problem is a quadratically constrained program (QCP).

*Proof.* The proof is included in Appendix C.6. $\qquad\square$

This surrogate yields a tractable convex restriction of the MIQCP.

**Assumption 4.7** (Bounded kernel). Let $\kappa : \Theta \times \Theta \to \mathbb{R}$ be a positive definite kernel with RKHS $\mathcal{H}$ and feature map $\phi : \Theta \to \mathcal{H}$. Assume there exists a constant $\kappa_0 > 0$ such that

$$\kappa(\theta, \theta) \leq \kappa_0^2, \qquad \forall\, \theta \in \Theta.$$

The Gaussian RBF kernel $\kappa(\theta, \theta') = \exp(-\gamma\|\theta - \theta'\|^2)$ satisfies this with $\kappa_0 = 1$, which is the kernel used throughout Experiment Section 6.

**Theorem 4.8** (Finite-sample coverage of the KDRRA ambiguity set). *Fix a shift index $k \in [K]$ and let $P_{\star,k}$ denote the unknown true distribution of the future classifier parameter $\tilde{\theta}_k$. Let $\theta_1, \ldots, \theta_{n_k} \overset{\text{iid}}{\sim} P_{\star,k}$, and define the empirical distribution $\hat{P}_{n_k} = \frac{1}{n_k}\sum_{i=1}^{n_k}\delta_{\theta_i}$ with empirical kernel mean embedding $\hat{\mu}_{n_k} = \frac{1}{n_k}\sum_{i=1}^{n_k}\phi(\theta_i)$.*

*Under Assumption 4.7, for every $\delta \in (0,1)$, the following concentration bound holds:*

$$\mathbb{P}(\|\hat{\mu}_{n_k} - \mu_{\star,k}\|_{\mathcal{H}} \leq \varepsilon_{n_k}(\delta)) \geq 1 - \delta,$$

*where $\mu_{\star,k} = \mathbb{E}_{\theta \sim P_{\star,k}}[\phi(\theta)]$ is the true kernel mean embedding and*

$$\varepsilon_{n_k}(\delta) := \frac{\kappa_0}{\sqrt{n_k}} + \kappa_0 \sqrt{\frac{2\log(1/\delta)}{n_k}}.$$

*Equivalently, choosing the ambiguity radius $\varepsilon_k = \varepsilon_{n_k}(\delta)$ guarantees:*

$$\mathbb{P}\left( P_{\star,k} \in \mathcal{K}_{\varepsilon_k}(\hat{P}_{n_k}) \right) \geq 1 - \delta,$$

*so that the empirical MMD ball centered at $\hat{P}_{n_k}$ is a statistically valid ambiguity set for the unknown future parameter distribution $P_{\star,k}$ with confidence $1 - \delta$.*

*Proof.* The proof follows the standard RKHS mean-embedding concentration strategy; see, e.g., Smola et al. (2007); Gretton et al. (2012); Muandet et al. (2017). For completeness, we provide the complete proof in Appendix C.1.

$\qquad\square$

**Corollary 4.9** (Uniform coverage across multiple shifts). Suppose Theorem 4.8 holds for each shift $k \in [K]$ with sample size $n_k$. Define

$$\varepsilon_k = \frac{\kappa_0}{\sqrt{n_k}} + \kappa_0 \sqrt{\frac{2\log(K/\delta)}{n_k}}.$$

Then

$$\mathbb{P}\left( P_{\star,k} \in \mathcal{K}_{\varepsilon_k}(\hat{P}_{n_k}), \ \forall k \in [K] \right) \geq 1 - \delta.$$

*Proof.* Applying Theorem 4.8 to each $k \in [K]$ with confidence level $\delta/K$ gives

$$\mathbb{P}\left(P_{\star,k} \notin \mathcal{K}_{\varepsilon_k}(\hat{P}_{n_k})\right) \leq \frac{\delta}{K}.$$

By the union bound,

$$\mathbb{P}\left(\exists\, k \in [K] : P_{\star,k} \notin \mathcal{K}_{\varepsilon_k}(\hat{P}_{n_k})\right) \leq \sum_{k=1}^{K} \frac{\delta}{K} = \delta.$$

Taking complements yields the result. $\qquad\qquad\square$

Although the RKHS-based ambiguity set provides a flexible nonparametric framework for modeling uncertainty, its effectiveness depends critically on the choice of the ambiguity-set center. In the above formulation, this center is taken as the empirical distribution, whose kernel mean embedding captures sampling variability but can become unstable under limited data or distribution shift, often necessitating a larger radius $\varepsilon$ and leading to overly conservative recourse. To mitigate this issue, we seek a reference distribution that remains data-driven while being more stable and less sensitive to sampling noise or moderate model misspecification. This motivates a Bayesian perspective, in which uncertainty is incorporated directly into the reference distribution via posterior predictive averaging. We therefore replace the empirical center with a Bayesian posterior predictive distribution and develop the resulting Bayesian-centered ambiguity set construction in the next section, along with the reformulation of the BKDRRA problem.

## 5 Bayesian Kernel Distributionally Robust Recourse Action

This section presents BKDRRA, which replaces the ambiguity-set center with a posterior-bootstrap Bayesian predictive distribution, and derives SOCP formulations via dual and representer-theorem reductions, along with finite-sample coverage.

### 5.1 Bayesian-Centered Ambiguity Set Construction

In MMD-based DRO for recourse, the ambiguity set is centered at an estimator of the data-generating process. A common choice is the empirical distribution $\widehat{P}_n$, which is completely data-driven Netessine et al. (2019); Staib & Jegelka (2019); Zhu et al. (2021). When reliable prior knowledge or structural assumptions are available, one may instead use a model-based estimator $\widetilde{P}$ Iyengar et al. (2023); Michel et al. (2021). However, these choices have complementary vulnerabilities: $\widehat{P}_n$ is sensitive to sampling error, whereas $\widetilde{P}$ is sensitive to model misspecification. To ensure coverage of the true test distribution, one often increases the MMD radius $\varepsilon$, but this leads to excessive conservatism and inflated recourse cost.

We define the ambiguity-set center as a *Bayesian posterior predictive* distribution $P_n^{\mathrm{pred}}$, constructed via nonparametric learning (NPL) Rubin (1981); Lyddon et al. (2018). This choice implicitly averages over plausible data-generating models under the posterior, thereby internalizing both sampling variability and moderate misspecification into the reference distribution itself. We therefore define the ambiguity set as an RKHS MMD ball centered at the posterior predictive distribution $P_n^{\mathrm{pred}}$.

which replaces empirical- or model-centered balls in the recourse formulation.

### Construction of the Bayesian Posterior Predictive

Let $\mathcal{D}_n = \{z_i\}_{i=1}^n$ denote the observed dataset.[1] Throughout this subsection, we denote by $\varphi \in \Phi$ the parameter indexing a family of candidate *distributions* $\{\mathbb{P}_\varphi : \varphi \in \Phi\}$ that are used exclusively for constructing the ambiguity-set center. This choice avoids any notational conflict with the symbol $\tilde{\theta}$, which elsewhere in the paper represents uncertain *recourse model parameters*.

---

[1]Depending on context, the observations $z_i$ may correspond to input-output pairs, model residuals, covariates, or generic samples from the underlying data-generating mechanism. This should be made explicit when integrating the predictive distribution into the recourse model.

To capture uncertainty in the empirical distribution without imposing a parametric likelihood, we adopt the posterior–bootstrap (likelihood–free Bayesian bootstrap) approach. Define a random weighted empirical measure

$$G^{(b)} \; = \; \sum_{i=1}^{n} w_i^{(b)} \, \delta_{z_i}, \qquad \boldsymbol{w}^{(b)} \sim \mathrm{Dirichlet}(1, \ldots, 1), \tag{12}$$

which provides a nonparametric Bayesian draw of the data-generating distribution. Each weight vector $\boldsymbol{w}^{(b)}$ induces a distinct perturbation of the empirical measure, thereby generating a distributional sample that reflects posterior uncertainty under the Bayesian bootstrap.

For each bootstrap sample $b = 1, \ldots, B$, we fit the model distribution $\mathbb{P}_\varphi$ to the randomly weighted empirical measure $G^{(b)}$ via a MMD minimization step:

$$\varphi^{(b)} \; \in \; \arg\min_{\varphi \in \Phi} \left\{ \mathrm{D}_\kappa^2 \left( \mathbb{P}_\varphi, G^{(b)} \right) \right\}, \tag{13}$$

where $\mathrm{D}_\kappa$ denotes the MMD associated with the reproducing kernel $\kappa$. This step effectively computes the best-fitting member of the model class $\{\mathbb{P}_\varphi\}$ for each posterior–bootstrap realization of the data distribution.

**Bayesian posterior predictive.** The posterior predictive distribution is defined as the posterior expectation of the model distribution i. e.,

$$\mathbb{P}_n^{\mathrm{pred}} \; = \; \mathbb{E}[\mathbb{P}_\varphi \,|\, \mathcal{D}_n] \; \approx \; \frac{1}{B} \sum_{b=1}^{B} \mathbb{P}_{\varphi^{(b)}}, \tag{14}$$

where the approximation arises from Monte Carlo averaging over the posterior–bootstrap samples. The mixture representation above aggregates the ensemble of fitted distributions $\mathbb{P}_{\varphi^{(b)}}$, thereby incorporating both parameter uncertainty (via $\varphi$) and model uncertainty (via the random bootstrap measures $G^{(b)}$). Consequently, $\mathbb{P}_n^{\mathrm{pred}}$ serves as a robust and data-adaptive estimate of the nominal center for our ambiguity set construction.

**The ambiguity set is centered at the posterior predictive.** We construct the MMD-based ambiguity set around the posterior predictive distribution as:

$$\mathcal{B}_\varepsilon \left( \mathbb{P}_n^{\mathrm{pred}} \right) \; = \; \left\{ P : \mathrm{D}_\kappa \left( \mathbb{P}, \, \mathbb{P}_n^{\mathrm{pred}} \right) \le \varepsilon \right\}. \tag{15}$$

Unlike empirical-centered ambiguity sets (which capture sampling variability only), this posterior-predictive-centered ball accounts for both posterior dispersion and model-fitting uncertainty induced by the Bayesian bootstrap. In the subsequent recourse formulation, all worst-case expectations are taken over the distributions in $\mathcal{B}_\varepsilon \left( P_n^{\mathrm{pred}} \right)$, thereby ensuring that both statistical and model-based uncertainties are rigorously incorporated into the Bayesian DRO problem 16.

This substitution does not alter the tractable MMD dual and representer-based reformulation. At the same time, the uncertainty-aware center typically permits a smaller ambiguity radius $\varepsilon$ while still covering the true data-generating distribution.

## 5.2 Bayesian Kernel Distributionally Robust Recourse Formulation

We now study the distributionally robust counterpart of the recourse problem 2, where the uncertainty in the future classifier parameters is modeled through the kernel-based Bayesian ambiguity set 15. For every shift index $k \in [K]$, the corresponding worst-case recourse optimization reads:

$$\begin{aligned} \min_{x \in X} \quad & d(x, x_0), \\ \text{s.t.} \quad & \sup_{\mathbb{P} \in \mathcal{B}_{\varepsilon_k}^k \left( P_n^{\mathrm{pred}} \right)} \mathbb{P} \left( C_{\tilde{\theta}_k}(x) \le 0 \right) \; \le \; \delta, \qquad \forall \, k \in [K]. \end{aligned} \tag{16}$$

The uncertainty enters only through the violation probability inside the constraint. Therefore, for any fixed action $x$ and shift $k$, we isolate the worst-case probability:

$$\sup_{\mathbb{P} \in \mathcal{B}^k_{\varepsilon_k}\left(\mathbb{P}^{\mathrm{pred}}_n\right)} \mathbb{P}\left(C_{\tilde{\theta}_k}(x) \leq 0\right). \tag{17}$$

Direct evaluation of 17 is intractable due to the infinite-dimensional nature of the ambiguity set. To obtain a computable upper bound, we leverage structural properties of MMD-based ambiguity sets in RKHSs. In particular, their dual representation allows the worst-case probability to be bounded by an optimization problem over RKHS functions that upper bound the violation indicator 20.

**Kernel Discrepancy**

The kernel discrepancy (or MMD) associated with the RKHS $\mathcal{H}$ is also given by Gretton et al. (2012)

$$D_\kappa(\mathbb{P}, \mathbb{Q}) = \sup_{\|h\|_{\mathcal{H}} \leq 1} \left(\mathbb{E}_{\mathbb{P}}[h] - \mathbb{E}_{\mathbb{Q}}[h]\right), \tag{18}$$

and the ambiguity set $\mathcal{B}^k_{\varepsilon_k}(\mathbb{P}^{\mathrm{pred}}_n)$ contains all distributions whose kernel discrepancy from the predictive distribution $\mathbb{P}^{\mathrm{pred}}_n$ does not exceed $\varepsilon_k$.

**Proposition 5.1.** *For any measurable function $f$ dominated by some $h \in \mathcal{H}$ (i. e. $f \leq h$ pointwise), the worst-case expectation satisfies*

$$\sup_{\mathbb{P} \in \mathcal{B}^k_{\varepsilon_k}(\mathbb{P}^{\mathrm{pred}}_n)} \mathbb{E}_{\mathbb{P}}[f] \;\leq\; \inf_{h \in \mathcal{H}, \, f \leq h} \left\{\mathbb{E}_{\mathbb{P}^{\mathrm{pred}}_n}[h] + \varepsilon_k \|h\|_{\mathcal{H}}\right\}. \tag{19}$$

*Proof.* For any $\mathbb{P} \in \mathcal{B}^k_{\varepsilon_k}(\mathbb{P}^{\mathrm{pred}}_n)$ and any $h \in \mathcal{H}$ with $f \leq h$ pointwise,

$$\mathbb{E}_{\mathbb{P}}[f] \leq \mathbb{E}_{\mathbb{P}}[h] = \mathbb{E}_{\mathbb{P}^{\mathrm{pred}}_n}[h] + \left(\mathbb{E}_{\mathbb{P}}[h] - \mathbb{E}_{\mathbb{P}^{\mathrm{pred}}_n}[h]\right).$$

By definition of $D_\kappa(\cdot, \cdot)$, $\mathbb{E}_{\mathbb{P}}[h] - \mathbb{E}_{\mathbb{P}^{\mathrm{pred}}_n}[h] \leq D_\kappa(\mathbb{P}, \mathbb{P}^{\mathrm{pred}}_n) \|h\|_{\mathcal{H}} \leq \varepsilon_k \|h\|_{\mathcal{H}}$. Thus $\mathbb{E}_{\mathbb{P}}[f] \leq \mathbb{E}_{\mathbb{P}^{\mathrm{pred}}_n}[h] + \varepsilon_k \|h\|_{\mathcal{H}}$ for every feasible $h$, and taking $\sup_{\mathbb{P}}$ on the left and then $\inf_h$ on the right yields the claim. For detailed proof see Appendix C.7. $\qquad\square$

**Lemma 5.2.** *For the classifier misclassification indicator associated with action $x$, defined as*

$$f_x(\tilde{\theta}) = \mathbf{1}\{C_{\tilde{\theta}}(x) \leq 0\} = \mathbf{1}\{\tilde{\theta}^\top x \leq 0\}, \tag{20}$$

*the worst-case violation probability in 17 admits the upper bound:*

$$\sup_{\mathbb{P} \in \mathcal{B}^k_{\varepsilon_k}(\mathbb{P}^{\mathrm{pred}}_n)} \mathbb{P}\left(\tilde{\theta}^\top x \leq 0\right) = \sup_{\mathbb{P} \in \mathcal{B}^k_{\varepsilon_k}(\mathbb{P}^{\mathrm{pred}}_n)} \mathbb{E}_{\tilde{\theta} \sim \mathbb{P}}\left[f_x(\tilde{\theta})\right] \leq \inf_{h \in \mathcal{H}, \, f_x \leq h} \left\{\mathbb{E}_{\mathbb{P}^{\mathrm{pred}}_n}[h] + \varepsilon_k \|h\|_{\mathcal{H}}\right\}. \tag{21}$$

*Proof.* By applying a standard probability identity, we have

$$\mathbb{P}\left(\tilde{\theta}^\top x \leq 0\right) = \mathbb{E}_{\tilde{\theta} \sim \mathbb{P}}\left[\mathbf{1}\{\tilde{\theta}^\top x \leq 0\}\right].$$

Applying Proposition 5.1 with $f = f_x$ yields the result. $\qquad\square$

**Lemma 5.3.** *For each $(\kappa, x)$, there exists an optimizer of Lemma 5.2 of the form*

$$h^*_{\kappa,x}(\tilde{\theta}) = \sum_{j=1}^{J} \alpha_j \, \kappa(\tilde{\theta}, \xi_j),$$

*where $\{\xi_j\}_{j=1}^{J}$ is a finite set of kernel centers drawn from the support of the reference distribution and $\alpha_j \in \mathbb{R}$ are coefficients.*

*Proof.* This follows from a standard representer theorem argument for RKHS optimization. See complete proof in Appendix C.8. $\qquad\square$

**Choice of Kernel Sites.** To approximate the optimal envelope function $h^*_{\kappa,x}$ using a finite set of kernel evaluations, we must specify the locations of the kernel centers $\{\xi_j\}_{j=1}^J$. A natural choice is to include samples drawn from the posterior-predictive distribution:

$$\text{Posterior sites:} \qquad \{\tilde{\theta}_k^{(b)}\}_{b=1}^{B_k} \sim \mathbb{P}_n^{\text{pred}},$$

which represent plausible realizations of the future classifier under shift $k$.

**Near-Boundary Sites.** In addition to posterior samples, we also include points positioned near the decision boundary $\{\tilde{\theta} : \tilde{\theta}^\top x = 0\}$. Let $\{\xi_m\}_{m=1}^M$ denote such boundary-adjacent sites, selected so that any feasible upper envelope $h$ satisfies

$$h(\xi_m) \geq 1 \quad \text{whenever} \quad \tilde{\theta}^\top x \leq 0.$$

With these kernel centers, every candidate envelope admits the expansion,

$$h(\tilde{\theta}) = \sum_{j=1}^J \alpha_j \, \kappa\big(\tilde{\theta}, \xi_j\big),$$

and its RKHS norm takes the quadratic form

$$\|h\|_{\mathcal{H}}^2 = \boldsymbol{\alpha}^\top \mathcal{K} \boldsymbol{\alpha},$$

where $\mathcal{K}$ is the Gram matrix with entries $\mathcal{K}_{ij} = \kappa(\xi_i, \xi_j)$.

**Empirical Approximation of the Predictive Expectation.** The expectation of $h$ under the predictive distribution can be approximated empirically using the posterior sites:

$$\mathbb{E}_{P_n^{\text{pred}}}[h] \; \approx \; \frac{1}{B_k} \sum_{b=1}^{B_k} h\big(\tilde{\theta}_k^{(b)}\big) = \frac{1}{B_k} \mathbf{1}^\top \mathcal{K}_{B_k, J} \, \boldsymbol{\alpha},$$

where $\mathcal{K}_{B_k, J}$ is the cross-kernel matrix whose $(b,j)$ entry is $\kappa\big(\tilde{\theta}_k^{(b)}, \xi_j\big)$.

**Boundary Dominance Constraints.** The requirement $f_x \leq h$ enforces that $h(\xi_m) \geq 1$ for all boundary-adjacent sites, which yields the linear inequalities

$$\mathcal{K}(\xi_m)^\top \boldsymbol{\alpha} \geq 1, \qquad m = 1, \ldots, M,$$

where $\mathcal{K}(\xi_m)$ denotes the vector of kernel evaluations $\kappa(\xi_m, \xi_j)$.

**Finite-Dimensional Convex Program.** Substituting the empirical expectation and the RKHS norm into the dual representation of the worst-case violation probability for shift $k$ yields a finite dimensional convex optimization problem in the coefficient vector $\boldsymbol{\alpha}$:

$$\min_{\boldsymbol{\alpha} \in \mathbb{R}^J} \quad \frac{1}{B_k} \mathbf{1}^\top \mathcal{K}_{B_k, J} \, \boldsymbol{\alpha} \; + \; \varepsilon_k \sqrt{\boldsymbol{\alpha}^\top \mathcal{K} \boldsymbol{\alpha}} \tag{22}$$
$$\text{s.t.} \quad \mathcal{K}(\xi_m)^\top \boldsymbol{\alpha} \geq 1, \qquad m = 1, \ldots, M.$$

The optimal value of 22 provides an upper bound on $\sup_{\mathbb{P} \in \mathcal{B}_{\varepsilon_k}^k(\mathbb{P}_n^{\text{pred}})} \mathbb{P}(C_{\tilde{\theta}_k}(x) \leq 0)$.

**Proposition 5.4.** *Problem 22 is equivalent to a second-order cone program.*

*Proof.* Consider the optimization problem 22

$$\min_{\boldsymbol{\alpha} \in \mathbb{R}^J} \quad \frac{1}{B_k} \mathbf{1}^\top \mathcal{K}_{B_k, J} \, \boldsymbol{\alpha} \; + \; \varepsilon_k \sqrt{\boldsymbol{\alpha}^\top \mathcal{K} \boldsymbol{\alpha}}$$
$$\text{s.t.} \quad \mathcal{K}(\xi_m)^\top \boldsymbol{\alpha} \geq 1, \qquad m = 1, \ldots, M.$$

Introduce an auxiliary scalar variable $t \geq 0$ and rewrite the epigraph reformulation problem equivalently as

$$
\begin{aligned}
\min_{\boldsymbol{\alpha} \in \mathbb{R}^J, \, t \in \mathbb{R}} \quad & \frac{1}{B_k} \mathbf{1}^\top \mathcal{K}_{B_k, J} \, \boldsymbol{\alpha} \, + \, \varepsilon_k \, t \\
\text{s.t.} \quad & \sqrt{\boldsymbol{\alpha}^\top \mathcal{K} \boldsymbol{\alpha}} \leq t, \\
& \mathcal{K}(\xi_m)^\top \boldsymbol{\alpha} \geq 1, \qquad m = 1, \ldots, M, \\
& t \geq 0.
\end{aligned}
\tag{23}
$$

Because $\varepsilon_k \geq 0$, the objective function is nondecreasing in $t$. Hence, at any optimal solution of 23, the inequality $\sqrt{\boldsymbol{\alpha}^\top \mathcal{K} \boldsymbol{\alpha}} \leq t$ holds with equality. Therefore, problems 22 and 23 are equivalent.

Since $\mathcal{K}$ is a Gram matrix, it is symmetric positive semidefinite. Thus there exists a matrix $R$ such that

$$
\mathcal{K} = R^\top R.
$$

It follows that

$$
\boldsymbol{\alpha}^\top \mathcal{K} \boldsymbol{\alpha} = \boldsymbol{\alpha}^\top R^\top R \boldsymbol{\alpha} = \|R \boldsymbol{\alpha}\|_2^2.
$$

Hence the constraint $\sqrt{\boldsymbol{\alpha}^\top \mathcal{K} \boldsymbol{\alpha}} \leq t$ is equivalent to

$$
\|R \boldsymbol{\alpha}\|_2 \leq t,
\tag{24}
$$

which defines a second-order cone. The remaining constraints are affine. Therefore, problem 23 is a second-order cone program (SOCP).

Since 23 is equivalent to 22, the latter admits an exact SOCP reformulation. $\qquad \square$

**Theorem 5.5.** *Consider the Bayesian kernel distributionally robust recourse problem 16 with ambiguity sets $\mathcal{B}_{\varepsilon_k}^k(P_n^{\mathrm{pred}})$ defined in 15. The BKDRRA problem admits the following joint finite-dimensional reformulation:*

$$
\begin{aligned}
\min_{\{\boldsymbol{\alpha}^{(k)}\}_{k=1}^K, \, x \in \mathcal{X}} \quad & d(x_0, x) \\
\text{s.t.} \quad & \frac{1}{B_k} \mathbf{1}^\top \mathcal{K}_{B_k, J}^{(k)} \, \boldsymbol{\alpha}^{(k)} + \varepsilon_k \sqrt{(\boldsymbol{\alpha}^{(k)})^\top \mathcal{K}^{(k)} \boldsymbol{\alpha}^{(k)}} \, \leq \, \delta, \qquad \forall \, k \in [K], \\
& \mathcal{K}^{(k)}(\xi_m^{(k)})^\top \boldsymbol{\alpha}^{(k)} \geq 1, \qquad m = 1, \ldots, M_k, \; \forall \, k \in [K].
\end{aligned}
\tag{25}
$$

*Here, for each distributional shift index $k \in [K]$: $\mathcal{K}^{(k)}$ is the Gram matrix with entries $\mathcal{K}_{ij}^{(k)} = \kappa\left(\xi_i^{(k)}, \xi_j^{(k)}\right)$, $\mathcal{K}_{B_k, J}^{(k)}$ is the cross-kernel matrix with entries $\left(\mathcal{K}_{B_k, J}^{(k)}\right)_{bj} = \kappa\left(\tilde{\theta}_k^{(b)}, \xi_j^{(k)}\right)$, $\{\tilde{\theta}_k^{(b)}\}_{b=1}^{B_k} \sim \mathbb{P}_n^{\mathrm{pred}(k)}$ are posterior predictive samples, and $\{\xi_m^{(k)}\}_{m=1}^{M_k}$ are boundary dominance points used to enforce envelope validity.*

*Moreover, for each $k \in [K]$, the first constraint in 25 admits an equivalent second-order cone (SOC) representation.*

*Proof.* Fix $k \in [K]$ and $x \in X$. By Lemma 5.2, the worst-case probability 17 is upper bounded by an RKHS envelope problem over $h \in \mathcal{H}$ satisfying $f_x \leq h$. By Lemma 5.3, an optimizer admits a finite expansion $h(\tilde{\theta}) = \sum_{j=1}^J \alpha_j \kappa(\tilde{\theta}, \xi_j)$ over the chosen centers $\{\xi_j\}_{j=1}^J$. Approximating $\mathbb{E}_{\mathbb{P}_n^{\mathrm{pred}}}[h]$ by posterior sites $\{\tilde{\theta}_k^{(b)}\}_{b=1}^{B_k} \sim \mathbb{P}_n^{\mathrm{pred}}$ and enforcing $f_x \leq h$ via boundary dominance constraints yields 22. Imposing the resulting bound to be at most $\delta$ for all $k \in [K]$ and minimizing $d(x_0, x)$ over $x \in X$ gives 25. The SOCP representability follows from Proposition 5.4. $\qquad \square$

The formulation 25 is a second-order cone program and can be solved efficiently using standard convex optimization software such as CVX Grant et al. (2009).

**Theorem 5.6** (Finite-sample guarantee for BKDRRA posterior predictive embedding). *Let $D_n := \{\theta_1, \ldots, \theta_n\}$ denote the observed dataset used to construct the posterior predictive distribution $P_n^{\mathrm{pred}}$, which is used to center the BKDRRA ambiguity set. Let*

$$\tilde{\theta}^{(1)}, \ldots, \tilde{\theta}^{(B)} \overset{\text{iid}}{\sim} P_n^{\mathrm{pred}} \qquad \text{conditionally on } D_n.$$

*Define*

$$\hat{\mu}_B^{\mathrm{pred}} := \frac{1}{B} \sum_{b=1}^{B} \phi(\tilde{\theta}^{(b)}), \qquad \mu_n^{\mathrm{pred}} := \mathbb{E}\big[\phi(\tilde{\theta}) \mid D_n\big].$$

*Under Assumption 4.7, for every $\delta \in (0, 1)$,*

$$\mathbb{P}\left( \|\hat{\mu}_B^{\mathrm{pred}} - \mu_n^{\mathrm{pred}}\|_{\mathcal{H}} \leq \frac{\kappa_0}{\sqrt{B}} + \kappa_0 \sqrt{\frac{2 \log(1/\delta)}{B}} \;\middle|\; D_n \right) \geq 1 - \delta.$$

*Proof.* Conditional on $D_n$, the samples $\tilde{\theta}^{(1)}, \ldots, \tilde{\theta}^{(B)}$ are i.i.d. from $P_n^{\mathrm{pred}}$. Therefore, the proof is identical to that of Theorem 4.8, with $B$ in place of $n_k$, $\mu_n^{\mathrm{pred}}$ in place of $\mu_{\star,k}$, and $\hat{\mu}_B^{\mathrm{pred}}$ in place of $\hat{\mu}_{n_k}$. □

*Remark* 5.7. Both KDRRA and BKDRRA can be formulated as QCP or SOCP problems. For KDRRA, we employ the QCP formulation. This choice is motivated by the direct construction of the ambiguity set from empirical samples, which yields a tractable and tight optimization problem. In contrast, BKDRRA requires an additional computational layer to construct the posterior predictive distribution, substantially increasing problem complexity. To maintain computational scalability, we therefore adopt the SOCP formulation for BKDRRA. While a QCP formulation for BKDRRA could be derived analogously to the procedure in Section 4 (substituting the empirical distribution with samples from the posterior predictive distribution), the resulting optimization problem becomes prohibitively expensive in practice. For completeness, we provide an SOCP reformulation of KDRRA in Appendix B and empirically compare it with the QCP version. Due to the significantly higher computational cost, we do not include a corresponding QCP–SOCP comparison for BKDRRA.

# 6 Experiments

We evaluate our proposed KDRRA model 11 and BKDRRA model 25 against five strong baseline models: AR Ustun et al. (2019), Wachter Wachter et al. (2017), ROAR Upadhyay et al. (2021), DiRRAc Nguyen et al. (2023), and Gaussian DiRRAc Nguyen et al. (2023). Table 1 records the summary features of these baselines and our proposed models.

All experiments are conducted on four benchmark datasets obtained from the UCI Machine Learning Repository and Kaggle:

- **German Credit** Dua & Graff (2017): The dataset consists of 1000 loan applicant records with a binary target variable indicating creditworthiness. We use the corrected version of the dataset, in which inconsistencies in the Status attribute have been resolved. The input features include status and personal status (categorical), duration (in months), credit amount, and age. Categorical variables are transformed via one-hot encoding. After one-hot encoding, the feature dimension is 8.

- **Small Business Administration (SBA) Loans** Li et al. (2018): We use publicly available U.S. Small Business Administration (SBA) 1159 loan records with binary repayment outcomes, a benchmark dataset in prior work on robust and distributionally robust recourseNguyen et al. (2023); Upadhyay et al. (2021). We use 13 features: *Selected*, *UrbanRural*, *New*, *RealEstate*, and *Recession* (categorical), and *Term*, *NoEmp*, *CreateJob*, *RetainedJob*, *ChgOffPrinGr*, *GrAppv*, *SBA_Appv*, and *Portion* (numerical), resulting in a 19-dimensional input after one-hot encoding. To evaluate robustness under distribution shift, we induce temporal covariate shift by partitioning the data by loan

issuance year: loans issued during 1989–2006 define the source (training) distribution, while loans issued in subsequent years form shifted target distributions. This construction follows standard temporal shift protocols in robust learning.

- **Student Performance** Silva (2008): This dataset contains academic and demographic records of 649 secondary school students from two Portuguese schools. We formulate a binary classification task using the final grade $G_3$, labeling students with $G_3 < 12$ as fail and those with $G_3 \geq 12$ as pass. The input consists of nine features:*Age*, *Absences*, *G1*, and *G2* (numerical), and *Study time*, *Famsup*, *Higher*, *Internet*, and *Health* (categorical). Categorical variables are encoded using one-hot representations, resulting in 19-dimensional feature data. Distributional shift is modeled by separating the data by school identity, treating one school as the source distribution and the other as a shifted distribution.

- **Adult Income** Dua & Graff (2017): The dataset consists of 48,842 census records from the 1994 U.S. Census database with a binary target variable indicating whether an individual's annual income exceeds $50K. We formulate a binary classification task using the income label, where individuals earning $\leq 50K$ are treated as the unfavorable outcome and those earning $> 50K$ as the favorable outcome. The input features include demographic and socioeconomic attributes such as *Age*, *Education*, *EducationNum*, *WorkClass*, *MaritalStatus*, *Occupation*, *Relationship*, *CapitalGain*, *CapitalLoss*, *HoursPerWeek*, and *NativeCountry*. Categorical variables are encoded using one-hot representations, resulting in a 51-dimensional feature vector. To evaluate robustness under distribution shift, we construct a demographic covariate shift by partitioning the data according to age: individuals with Age $\leq 40$ define the source (training) distribution, while individuals with Age $> 40$ form the shifted target distribution.

To evaluate validity under distribution shift, we first partition each dataset into a source (original) distribution and a shifted target distribution, as described above. The source data is used to train the current classifier for which recourses are generated, while the shifted data is reserved for evaluating robustness under model perturbations.

To model uncertainty in the classifier parameters, we adopt an empirical sampling procedure. Specifically, we repeatedly draw random 80% subsamples from the original dataset and train a logistic regression model on each subsample. Repeating this process independently 100 times yields a collection of parameter realizations $\theta_1, \theta_2, \ldots, \theta_{100}$, which serve as empirical samples of the parameter distribution.

For evaluation under distribution shift, we repeatedly sample 20% subsets from the shifted dataset and train 100 corresponding classifiers.

During evaluation, we compute the following metrics to assess the robustness–cost trade-off of each method across a family of shifted distributions.

- **Validity** $M_1$: nominal validity under the base classifier.

- **Robust Validity** $M_2$: worst-case probability of positive classification under uncertainty.

- $l_1$ **Distance**: action cost in the feature space.

- $l_2$ **Distance**: Euclidean action cost.

- **Standard deviation**: variability across all test instances.

Define the per-classifier validity indicator for instance $i$ under classifier $j$ as

$$v_i^{(j)} \;=\; \mathbb{I}(C_{\theta^{(j)}}(x_i) = 1).$$

The $M_2$ *validity* (average fraction of shifted classifiers for which the recourse is valid, then averaged over instances) is

$$M_2 \;=\; \frac{1}{N}\sum_{i=1}^{N}\left(\frac{1}{J}\sum_{j=1}^{J} v_i^{(j)}\right) \;=\; \frac{1}{NJ}\sum_{i=1}^{N}\sum_{j=1}^{J}\mathbb{I}(C_{\theta^{(j)}}(x_i) = 1).$$

The $M_1$ *validity* (with respect to the original classifier) is

$$M_1 \;=\; \frac{1}{N}\sum_{i=1}^{N}\mathbb{I}(C_{\theta_0}(x_i)=1)\,.$$

For each instance $i$, define

$$\ell_1\text{-cost}_i \;=\; \|x_i - x_{0,i}\|_1, \qquad \ell_2\text{-cost}_i \;=\; \|x_i - x_{0,i}\|_2.$$

Report the average costs:

$$\ell_1 \;=\; \frac{1}{N}\sum_{i=1}^{N}\|x_i - x_{0,i}\|_1, \qquad \ell_2 \;=\; \frac{1}{N}\sum_{i=1}^{N}\|x_i - x_{0,i}\|_2.$$

**Standard deviation.** Let $N$ denote the number of test instances. Define $m_i^{(1)} = \mathbb{I}(C_{\theta_0}(x_i)=1)$. The standard deviation For $M_1$ validity is

$$\text{Std}(M_1) = \sqrt{\frac{1}{N}\sum_{i=1}^{N}\left(m_i^{(1)} - M_1\right)^2}.$$

Similarly, we compute the standard deviation for $M_2$, $l_1$, and $l_2$.

## 6.1 Results and Analysis

**Implementation Details.** All experiments are run on a Windows 11 Pro (64-bit) machine with 12 GB RAM and a 12th Gen Intel Core i7 (1.40 GHz) processor. The KDRRA QCP formulation is solved using Gurobi Gurobi Optimization, LLC (2021). The BKDRRA SOCP formulation is solved using CVXPY with the ECOS solver Domahidi et al. (2013) in Python. We use a Gaussian (RBF) kernel on the parameter space with kernel bandwidth fixed to 1 and ambiguity radius $\epsilon = 0.1$ across all experiments. For the KDRRA surrogate, the hinge loss $\psi(t) = \max\{1, 1 - t\}$ is used to replace the indicator function. In BKDRRA, we model $P_\varphi$ as a finite Gaussian mixture. Baseline methods follow the original implementations and hyperparameter settings reported in their respective works. In all experiments, we use $K = 1$.

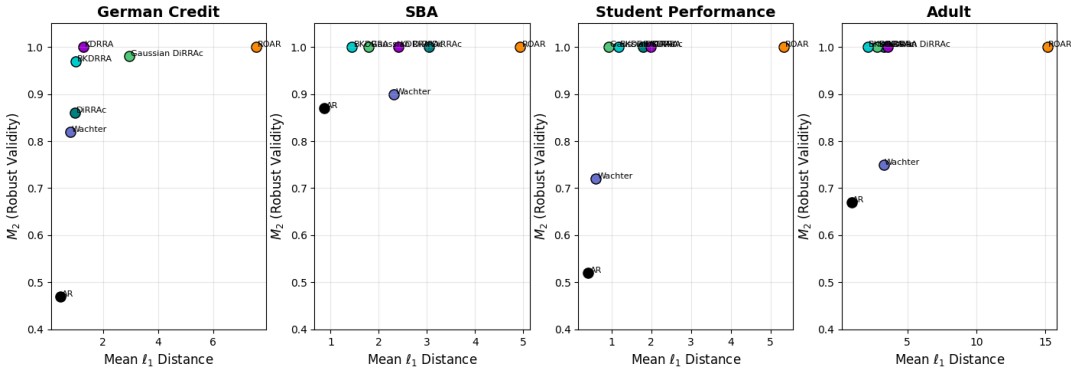

Figure 2: Scatter Plot: $M_2$ Validity vs $l_1$ Distance

Table 1: Comparison of major recourse frameworks.

| Method | Uncertainty Modeling | Ambiguity-Set Geometry | Conservatism Control | Optimization Structure |
|---|---|---|---|---|
| **AR / Wachter** | No uncertainty modeled; recourse computed for a fixed classifier. | No ambiguity set; single model instance. | Controlled by distance penalty and feasibility tolerance. | Gradient-based or convex optimization depending on classifier. |
| **ROAR** | Interval uncertainty on classifier parameters. | Hyper-rectangular (box-type) uncertainty set. | User-chosen interval width / sensitivity parameters. | Robust optimization reducible to linear or convex programs for linear/quadratic models. |
| **DiRRAc** | Distributional uncertainty via Wasserstein (Gelbrich) ambiguity sets. | Ellipsoidal / transport geometry induced by 2-Wasserstein metric. | Controlled by Wasserstein radius $\varepsilon$. | Min–max problem reformulated into convex programs (linear/SOCP settings). |
| **KDRRA (Proposed)** | Distributional uncertainty modeled using MMD in an RKHS. | RKHS ball around empirical kernel mean embedding; geometry set by kernel. | Controlled by MMD radius $\epsilon$ and kernel hyperparameters. | QCP reformulation using Gram-matrix representation of MMD constraints. |
| **BKDRRA (Proposed)** | Bayesian uncertainty via posterior predictive distribution (sampling + model uncertainty). | RKHS ball centered at posterior predictive embedding. | Controlled by $\epsilon$ and prior/posterior specification ( Bayesian bootstrap). | SOCP structure but with a tighter, less conservative ambiguity center. |

Table 2: Results on German Credit

| Method | $M_1$ | $M_2$ | $l_1$ | $l_2$ |
|---|---|---|---|---|
| AR | **1.00** $\pm$ 0.00 | 0.47 $\pm$ 0.05 | **0.46** $\pm$ 0.29 | **0.22** $\pm$ 0.14 |
| Wachter | **1.00** $\pm$ 0.00 | 0.82 $\pm$ 0.24 | 0.81 $\pm$ 0.51 | 0.41 $\pm$ 0.25 |
| ROAR | **1.00** $\pm$ 0.00 | **1.00** $\pm$ 0.00 | 7.57 $\pm$ 0.65 | 3.14 $\pm$ 0.27 |
| DiRRAc | **1.00** $\pm$ 0.00 | 0.86 $\pm$ 0.01 | 0.99 $\pm$ 0.94 | 0.70 $\pm$ 0.57 |
| Gaussian DiRRAc | **1.00** $\pm$ 0.00 | 0.98 $\pm$ 0.05 | 2.95 $\pm$ 0.47 | 2.04 $\pm$ 0.30 |
| KDRRA | *__1.00__* $\pm$ 0.00 | *__1.00__* $\pm$ 0.00 | 1.28 $\pm$ 0.80 | 0.66 $\pm$ 0.40 |
| BKDRRA | *__1.00__* $\pm$ 0.00 | 0.97 $\pm$ 0.05 | 1.01 $\pm$ 0.76 | 0.48 $\pm$ 0.36 |

Table 3: Results on SBA

| Method | $M_1$ | $M_2$ | $l_1$ | $l_2$ |
|---|---|---|---|---|
| AR | **1.00** $\pm$ 0.00 | 0.87 $\pm$ 0.03 | **0.87** $\pm$ 0.56 | **0.44** $\pm$ 0.28 |
| Wachter | **1.00** $\pm$ 0.00 | 0.90 $\pm$ 0.02 | 2.31 $\pm$ 2.39 | 0.77 $\pm$ 0.66 |
| ROAR | **1.00** $\pm$ 0.00 | **1.00** $\pm$ 0.00 | 4.93 $\pm$ 0.81 | 2.17 $\pm$ 0.34 |
| DiRRAc | **1.00** $\pm$ 0.00 | **1.00** $\pm$ 0.00 | 3.05 $\pm$ 0.55 | 1.70 $\pm$ 0.31 |
| Gaussian DiRRAc | **1.00** $\pm$ 0.00 | **1.00** $\pm$ 0.00 | 1.79 $\pm$ 0.54 | 1.01 $\pm$ 0.30 |
| KDRRA | ***1.00*** $\pm$ 0.00 | ***1.00*** $\pm$ 0.00 | 2.41 $\pm$ 0.56 | 1.26 $\pm$ 0.27 |
| BKDRRA | ***1.00*** $\pm$ 0.00 | ***1.00*** $\pm$ 0.00 | 1.44 $\pm$ 0.31 | 0.86 $\pm$ 0.18 |

Table 4: Results on Student Performance

| Method | $M_1$ | $M_2$ | $l_1$ | $l_2$ |
|---|---|---|---|---|
| AR | **1.00** $\pm$ 0.00 | 0.52 $\pm$ 0.11 | **0.41** $\pm$ 0.30 | **0.22** $\pm$ 0.16 |
| Wachter | **1.00** $\pm$ 0.00 | 0.72 $\pm$ 0.10 | 0.60 $\pm$ 0.43 | 0.30 $\pm$ 0.22 |
| ROAR | **1.00** $\pm$ 0.00 | **1.00** $\pm$ 0.00 | 5.33 $\pm$ 0.88 | 2.53 $\pm$ 0.41 |
| DiRRAc | **1.00** $\pm$ 0.00 | **1.00** $\pm$ 0.00 | 1.80 $\pm$ 0.26 | 1.33 $\pm$ 0.19 |
| Gaussian DiRRAc | **1.00** $\pm$ 0.00 | **1.00** $\pm$ 0.00 | 0.92 $\pm$ 0.24 | 0.68 $\pm$ 0.18 |
| KDRRA | ***1.00*** $\pm$ 0.00 | ***1.00*** $\pm$ 0.00 | 1.98 $\pm$ 0.21 | 0.98 $\pm$ 0.11 |
| BKDRRA | ***1.00*** $\pm$ 0.00 | ***1.00*** $\pm$ 0.00 | 1.17 $\pm$ 0.04 | 0.70 $\pm$ 0.02 |

Table 5: Results on Adult Dataset

| Method | $M_1$ | $M_2$ | $l_1$ | $l_2$ |
|---|---|---|---|---|
| AR | 1.00 $\pm$ 0.00 | 0.67 $\pm$ 0.03 | **0.98** $\pm$ 0.58 | **0.27** $\pm$ 0.16 |
| Wachter | **1.00** $\pm$ 0.00 | 0.75 $\pm$ 0.03 | 3.31 $\pm$ 1.82 | 0.50 $\pm$ 0.28 |
| ROAR | **1.00** $\pm$ 0.00 | **1.00** $\pm$ 0.00 | 15.12 $\pm$ 2.16 | 4.63 $\pm$ 0.60 |
| DiRRAc | **1.00** $\pm$ 0.00 | **1.00** $\pm$ 0.00 | 3.33 $\pm$ 0.19 | 1.09 $\pm$ 0.14 |
| Gaussian DiRRAc | **1.00** $\pm$ 0.00 | **1.00** $\pm$ 0.00 | 2.83 $\pm$ 0.55 | 1.01 $\pm$ 0.15 |
| KDRRA | 1.00 $\pm$ 0.00 | ***1.00*** $\pm$ 0.00 | 3.56 $\pm$ 0.89 | 1.97 $\pm$ 0.21 |
| BKDRRA | ***1.00*** $\pm$ 0.00 | ***1.00*** $\pm$ 0.00 | 2.14 $\pm$ 0.10 | 0.59 $\pm$ 0.01 |

For the German Credit dataset, KDRRA and BKDRRA demonstrate strong robustness performance. KDRRA attains 100% robust validity ($M_2 = 1.00$) with moderate action cost, while BKDRRA achieves slightly lower robustness (0.97) but significantly lower $l_1$ and $l_2$ distances, showing that Bayesian smoothing mitigates over-conservativeness. Among the baselines, AR provides the cheapest recourse but exhibits extremely poor robustness, and Wachter is cost-effective yet fails under parameter uncertainty. ROAR remains fully robust but highly conservative ($l_1 \approx 7.57$).

For the SBA loan dataset, which is more complex due to its high-dimensional categorical structure, all robust methods achieve perfect $M_2$ validity, except AR and Wachter, which fail under uncertainty. BKDRRA attains the lowest $l_1$ distance among all robust approaches, outperforming DiRRAc, Gaussian DiRRAc, and KDRRA, while KDRRA remains more conservative with larger $l_1$ and $l_2$ costs. Among the baselines, ROAR continues to be overly conservative, and Gaussian DiRRAc performs reasonably well but remains costlier than BKDRRA.

For the Student dataset, which is smaller and exhibits a different noise structure, all robust methods: DiRRAc, Gaussian DiRRAc, KDRRA, and BKDRRA, achieve 100% robust validity. BKDRRA provides the most efficient recourse, reducing the $l_1$ distance to approximately 1.17 and the $l_2$ distance to around 0.70, thereby achieving the best overall cost profile. KDRRA remains fully robust but is more conservative, with an $l_1$ cost of about 1.98. Among the baselines, AR and Wachter again fail to maintain robustness, while ROAR produces excessively large recourse costs.

For the Adult dataset, which is larger and contains a diverse set of demographic and socioeconomic attributes, all robust methods: DiRRAc, Gaussian DiRRAc, KDRRA, and BKDRRA, achieve 100% robust validity. BKDRRA provides the most efficient recourse, reducing the $l_1$ distance to approximately 2.14 and the $l_2$ distance to around 0.59, thereby achieving the best overall cost profile. KDRRA remains fully robust but is more conservative, with an $l_1$ cost of about 3.56. Among the baselines, AR and Wachter again fail to maintain robustness, with $M_2$ values of 0.67 and 0.75, respectively, while ROAR produces excessively large recourse costs ($l_1 \approx 15.12$).

In summary, our analysis reveals distinct performance profiles across the evaluated methods. KDRRA consistently achieves extremely high robustness (often with $M_2 = 1.0$) while maintaining competitive action cost. BKDRRA extends this performance by further reducing recourse costs while preserving near-perfect robustness, demonstrating the advantage of posterior predictive smoothing. In contrast, baseline methods show significant limitations: AR and Wachter collapse under realistic uncertainty conditions, ROAR remains excessively conservative for practical deployment, and Gaussian DiRRAc, while improving robustness, tends to overestimate uncertainty. Collectively, these results position BKDRRA as offering the optimal balance between cost efficiency and robustness, while KDRRA provides the strongest robustness guarantees.

## 6.2 Nonlinear Models via Local Linear Approximation

Following prior work Rawal et al. (2020); Upadhyay et al. (2021); Bui et al. (2022); Nguyen et al. (2023), we extend our framework to nonlinear classifiers using local linear approximations obtained via LIME ( Ribeiro et al. (2016)). Specifically, we train a multilayer perceptron (MLP) on each dataset and construct local surrogate linear models using LIME with 1000 perturbed samples. To reduce approximation variability, we generate 10 independent LIME models for each instance.

To capture model uncertainty, we construct empirical distributions over classifier parameters via repeated retraining on the original data. These samples are then used to build the ambiguity sets for KDRRA and BKDRRA. The LIME models are used solely to locally linearize the nonlinear classifier, enabling the application of linear recourse methods.

We generate recourse for 200 negatively classified test instances and evaluate performance using the same protocol as in the linear setting. Results in Table 6 show that robust methods improve $M_2$ validity at the cost of increased recourse magnitude. LIME-DiRRAc achieves the highest robustness across datasets, while KDRRA and BKDRRA attain comparable $M_2$ with substantially lower $l_1$ and $l_2$ costs. Notably, BKDRRA consistently provides the best robustness-cost trade-off, outperforming AR and Wachter while remaining significantly less conservative than ROAR.

## 7 Conclusion

In this work, we tackled the problem of generating reliable and robust algorithmic recourse under model-parameter uncertainty. Existing methods (AR, ROAR, Wachter) either ignore distributional uncertainty or become overly conservative, leading to invalid or costly recourse. To address this, we proposed two new frameworks: KDRRA and BKDRRA that explicitly model parameter uncertainty through kernel-based ambiguity sets.

KDRRA builds a nonparametric kernel ambiguity set over parameters to minimize worst-case misclassification while producing low-cost counter factuals. BKDRRA extends this by incorporating Bayesian posterior predictive sampling via posterior bootstrap, yielding smoother, uncertainty-aware ambiguity sets and more adaptive, cost-efficient recourse.

**Future Work.** An important direction is to develop recourse methods that are simultaneously robust to distributional shifts, adversarial perturbations, and training-time contamination, building on recent advances in joint DRO-ARO (Adversarial RO) formulations Selvi et al. (2024); Li et al. (2025). We also plan to explore richer and more structured ambiguity sets, including adversarial and data-dependent uncertainty models Bertolace et al. (2025). Finally, integrating recourse with representation learning (e.g., stable or

Table 6: Evaluation of Validity ($M_1$, $M_2$) and Cost ($l_1$, $l_2$) for Nonlinear Models

| Dataset | Method | $M_1$ | $M_2$ | $l_1$ | $l_2$ |
|---|---|---|---|---|---|
| German | LIME-AR | $0.30 \pm 0.40$ | $0.55 \pm 0.40$ | $0.50 \pm 0.15$ | $0.20 \pm 0.08$ |
| | Wachter | $\mathbf{1.00} \pm 0.00$ | $0.55 \pm 0.40$ | $\mathbf{0.20} \pm 0.20$ | $\mathbf{0.10} \pm 0.10$ |
| | LIME-ROAR | $0.40 \pm 0.45$ | $0.45 \pm 0.45$ | $5.20 \pm 0.30$ | $2.40 \pm 0.10$ |
| | LIME-DiRRAc | $0.60 \pm 0.45$ | $\mathbf{0.70} \pm 0.30$ | $2.20 \pm 1.00$ | $1.80 \pm 0.70$ |
| | LIME-Gaussian DiRRAc | $0.55 \pm 0.45$ | $0.65 \pm 0.35$ | $1.80 \pm 0.90$ | $1.40 \pm 0.70$ |
| | LIME-KDRRA | $0.55 \pm 0.45$ | $0.60 \pm 0.40$ | $0.70 \pm 0.20$ | $0.35 \pm 0.15$ |
| | LIME-BKDRRA | $0.60 \pm 0.45$ | $0.65 \pm 0.35$ | $0.45 \pm 0.12$ | $0.20 \pm 0.08$ |
| SBA | LIME-AR | $0.65 \pm 0.45$ | $0.80 \pm 0.30$ | $0.50 \pm 0.10$ | $0.20 \pm 0.05$ |
| | Wachter | $\mathbf{1.00} \pm 0.00$ | $0.65 \pm 0.40$ | $\mathbf{0.30} \pm 0.30$ | $\mathbf{0.12} \pm 0.10$ |
| | LIME-ROAR | $0.97 \pm 0.15$ | $\mathbf{1.00} \pm 0.00$ | $4.50 \pm 0.40$ | $1.50 \pm 0.15$ |
| | LIME-DiRRAc | $0.75 \pm 0.40$ | $0.80 \pm 0.35$ | $1.10 \pm 0.10$ | $1.05 \pm 0.10$ |
| | LIME-Gaussian DiRRAc | $0.80 \pm 0.35$ | $0.92 \pm 0.25$ | $0.60 \pm 0.25$ | $0.35 \pm 0.25$ |
| | LIME-KDRRA | $0.78 \pm 0.35$ | $0.92 \pm 0.20$ | $0.58 \pm 0.15$ | $0.24 \pm 0.08$ |
| | LIME-BKDRRA | $0.82 \pm 0.35$ | $0.94 \pm 0.18$ | $0.60 \pm 0.12$ | $0.25 \pm 0.06$ |
| Student | LIME-AR | $0.40 \pm 0.45$ | $0.30 \pm 0.35$ | $0.55 \pm 0.30$ | $0.20 \pm 0.10$ |
| | Wachter | $\mathbf{1.00} \pm 0.00$ | $0.40 \pm 0.40$ | $\mathbf{0.40} \pm 0.25$ | $\mathbf{0.20} \pm 0.15$ |
| | LIME-ROAR | $0.95 \pm 0.20$ | $0.90 \pm 0.30$ | $5.00 \pm 0.30$ | $1.80 \pm 0.30$ |
| | LIME-DiRRAc | $0.95 \pm 0.20$ | $\mathbf{0.95} \pm 0.25$ | $1.30 \pm 0.50$ | $1.20 \pm 0.40$ |
| | LIME-Gaussian DiRRAc | $0.65 \pm 0.45$ | $0.55 \pm 0.45$ | $0.70 \pm 0.60$ | $0.55 \pm 0.50$ |
| | LIME-KDRRA | $0.65 \pm 0.45$ | $0.60 \pm 0.45$ | $0.55 \pm 0.35$ | $0.35 \pm 0.25$ |
| | LIME-BKDRRA | $0.68 \pm 0.45$ | $0.59 \pm 0.40$ | $0.45 \pm 0.25$ | $0.28 \pm 0.15$ |
| Adult | LIME-AR | $0.50 \pm 0.45$ | $0.40 \pm 0.40$ | $0.60 \pm 0.05$ | $\mathbf{0.12} \pm 0.02$ |
| | Wachter | $\mathbf{1.00} \pm 0.00$ | $0.30 \pm 0.35$ | $1.20 \pm 0.70$ | $0.20 \pm 0.10$ |
| | LIME-ROAR | $0.99 \pm 0.10$ | $\mathbf{0.99} \pm 0.10$ | $10.00 \pm 0.20$ | $1.90 \pm 0.05$ |
| | LIME-DiRRAc | $0.99 \pm 0.10$ | $\mathbf{0.99} \pm 0.10$ | $1.05 \pm 0.05$ | $1.00 \pm 0.05$ |
| | LIME-Gaussian DiRRAc | $0.60 \pm 0.45$ | $0.50 \pm 0.40$ | $0.73 \pm 0.10$ | $0.15 \pm 0.05$ |
| | LIME-KDRRA | $0.75 \pm 0.40$ | $0.74 \pm 0.35$ | $0.70 \pm 0.10$ | $0.22 \pm 0.03$ |
| | LIME-BKDRRA | $0.80 \pm 0.35$ | $0.82 \pm 0.30$ | $0.79 \pm 0.10$ | $0.23 \pm 0.03$ |

invariant embeddings), as well as extending the framework to causal and streaming settings, constitutes a promising avenue for future research Mirzaei & Mathis (2024).

## Acknowledgments

The first author gratefully acknowledges the University Grants Commission (UGC), India, for providing the research fellowship that supported this work. The DST-FIST grant SR/FST/MS-1/2019/45 is acknowledged for the computing facility in the department at IIT Delhi.

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

# A    Appendix: Sensitivity and Computational Analysis

## A.1    Sensitivity and Computational Analysis

We conduct a sensitivity analysis to evaluate the effect of the ambiguity radius $\epsilon$ and the kernel bandwidth $\gamma$ on the robustness and cost of the proposed methods. For the radius analysis, we vary $\epsilon \in \{0, 0.05, 0.1, 0.2\}$ while fixing the kernel bandwidth to $\gamma = 1$, and measure the resulting robust validity $M_2$ and recourse cost $l_1$. As shown in Figures 3 and 4, increasing $\epsilon$ enlarges the uncertainty set, which improves robustness but generally leads to higher recourse cost. Across all datasets, KDRRA quickly achieves high or perfect robust validity as $\epsilon$ increases, whereas BKDRRA attains similar robustness while maintaining consistently lower recourse cost due to the posterior predictive smoothing of the uncertainty distribution.

We further analyze the sensitivity with respect to the kernel bandwidth $\gamma \in \{0.5, 1, 2, 5\}$ while fixing $\epsilon = 0.1$. The results in Figure 3 indicate that moderate bandwidth values yield the best robustness performance, whereas very small or large bandwidths slightly reduce robustness due to under- or over-smoothing of the kernel representation. As illustrated in Figure 4, the corresponding recourse costs remain relatively stable across a reasonable range of bandwidth values. Empirically, $\gamma = 1$ provides the most stable trade-off between robustness and cost across all datasets. Overall, these results demonstrate that the proposed methods are relatively robust to variations in these hyperparameters, with BKDRRA consistently achieving a favorable balance between robustness and recourse cost.

## A.2    Computational Runtime Analysis

We report the average runtime per instance for all methods across datasets in Table 7. Non-robust baselines such as AR and Wachter are computationally efficient, while robust methods incur higher cost due to the underlying optimization. Among the robust approaches, ROAR and DiRRAc are comparatively slower, whereas BKDRRA achieves improved efficiency due to its SOCP formulation. KDRRA exhibits moderate runtime, reflecting the trade-off between tighter robustness guarantees and computational cost. Overall, these results highlight the scalability–robustness trade-off inherent in distributionally robust recourse methods.

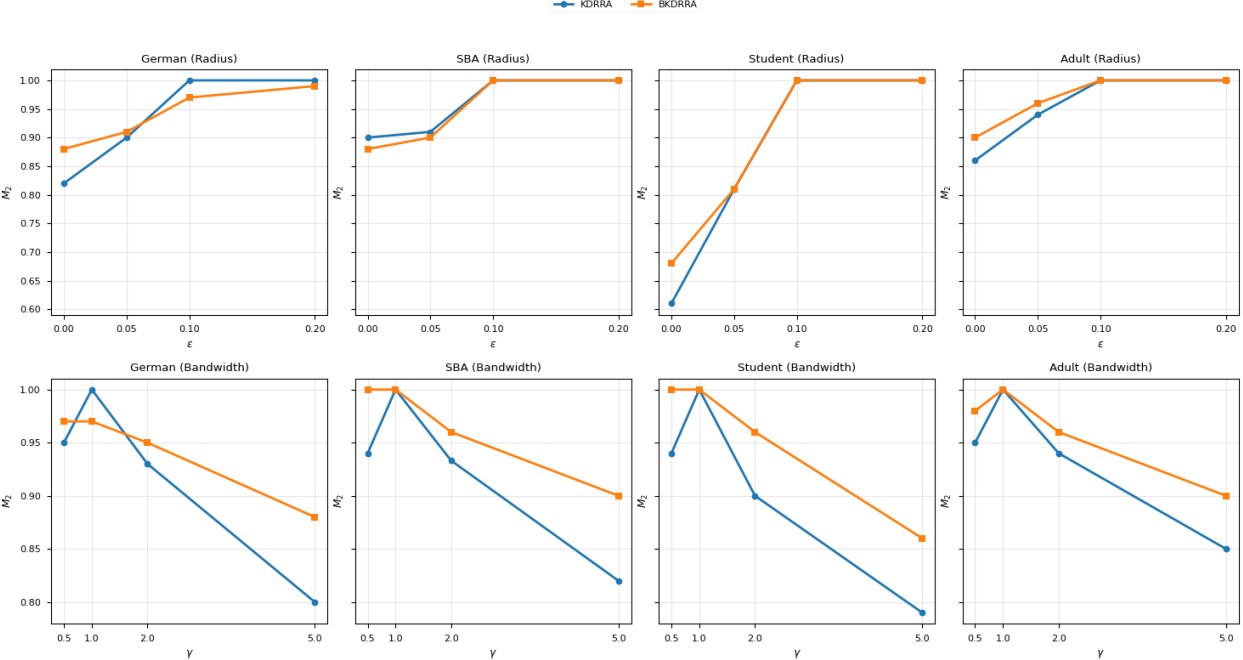

Figure 3: Sensitivity of robust validity $M_2$ with respect to ambiguity radius $\epsilon$ and kernel bandwidth $\gamma$.

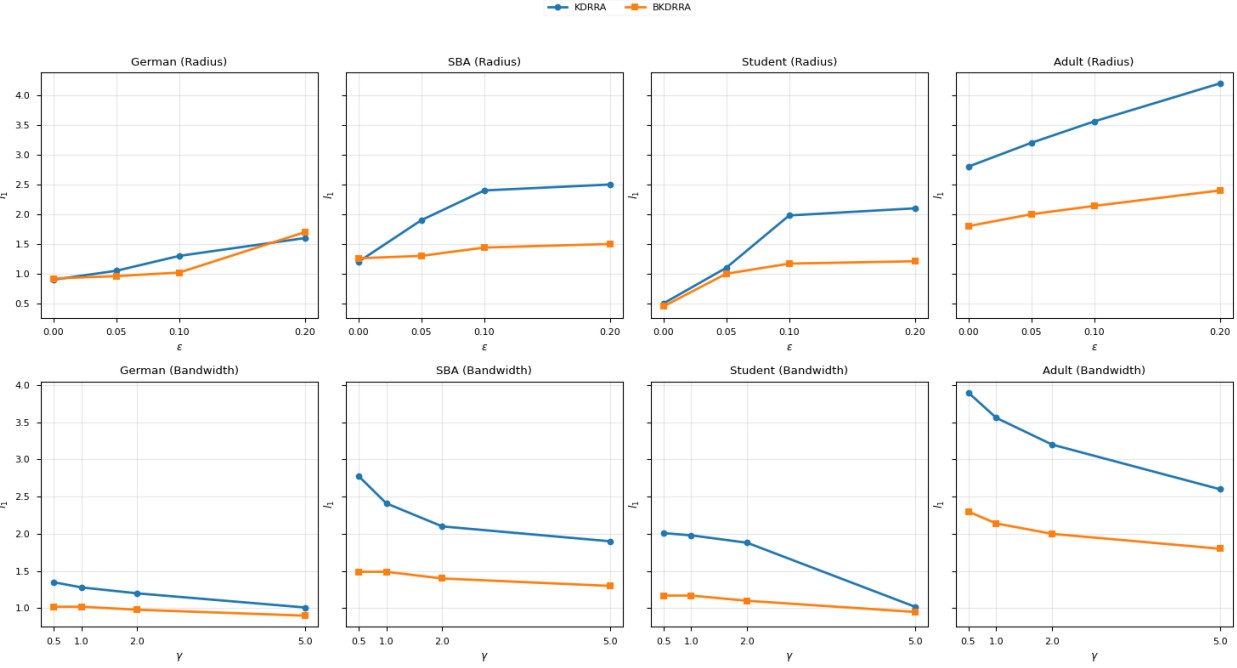

Figure 4: Sensitivity of recourse cost $l_1$ with respect to ambiguity radius $\epsilon$ and kernel bandwidth $\gamma$.

Table 7: Average runtime (seconds per instance) across datasets and methods

| Method | German | SBA | Student | Adult |
|---|---|---|---|---|
| AR | 0.027 | 0.046 | 0.038 | 0.10 |
| Wachter | 0.006 | 0.011 | 0.006 | 0.05 |
| ROAR | 0.38 | 0.31 | 0.399 | 5.10 |
| DiRRAc | 0.40 | 0.531 | 0.344 | 4.00 |
| Gauss-DiRRAc | 0.29 | 0.29 | 0.28 | 3.89 |
| KDRRA | 0.32 | 0.42 | 0.45 | 6.12 |
| BKDRRA | 0.11 | 0.15 | 0.154 | 1.13 |

# B  Appendix: Alternative SOCP Formulation of KDRRA

## B.1  SOCP Formulation

In this section, we reformulate the KDRRA problem using the same RKHS-based methodology employed in the BKDRRA formulation. The key difference is that the ambiguity set is centered at the empirical distribution of classifier parameters rather than the posterior predictive distribution.

Consider the robust recourse problem

$$\min_{x \in \mathcal{X}} d(x, x_0)$$

subject to

$$\sup_{\mathbb{P} \in \mathcal{P}_k} \mathbb{P}\big(C_{\tilde{\theta}_k}(x) \leq 0\big) \leq \delta, \quad \forall\, k \in [K].$$

**Step 1: Indicator Representation.**  Define the violation function

$$f_x(\tilde{\theta}) = \mathbf{1}\{C_{\tilde{\theta}}(x) \leq 0\}.$$

**Step 2: RKHS Upper Bound.**  Using the RKHS envelope upper bound 5.1, we obtain

$$\sup_{\mathbb{P} \in \mathcal{P}_k} \mathbb{E}_{\mathbb{P}}[f_x] \leq \inf_{h \in \mathcal{H},\, f_x \leq h} \{E_{\mathbb{P}_k}[h] + \epsilon_k \|h\|_{\mathcal{H}}\}.$$

**Step 3: Empirical Approximation.**  Let $\{\beta_i^k\}_{i=1}^{N_k}$ denote the collected samples of $\tilde{\theta}_k$. Then,

$$\mathbb{E}_{\mathbb{P}_k}[h] = \frac{1}{N_k} \sum_{i=1}^{N_k} h(\beta_i^k).$$

**Step 4: Representer Theorem.**  By the representer theorem, the optimal $h$ admits the form

$$h(\tilde{\theta}) = \sum_{j=1}^{J} \alpha_j^k \, \kappa(\tilde{\theta}, \tilde{\theta}_j^k).$$

Thus,

$$\|h\|_{\mathcal{H}}^2 = \alpha^{k\top} \mathcal{K}^{(k)} \alpha^k.$$

**Step 5: Substitution.**  Substituting into the objective, we obtain

$$\frac{1}{N_k} \sum_{i=1}^{N_k} \sum_{j=1}^{J} \alpha_j^k \, \kappa(\beta_i^k, \tilde{\theta}_j^k) + \epsilon_k \sqrt{\alpha^{k\top} \mathcal{K}^{(k)} \alpha^k}.$$

This can be written compactly as

$$\frac{1}{N_k} \mathbf{1}^\top \mathcal{K}_{N_k, J}^{(k)} \alpha^k + \epsilon_k \sqrt{\alpha^{k\top} \mathcal{K}^{(k)} \alpha^k}.$$

**Step 6: Boundary Dominance Constraint.** To ensure $f_x \leq h$, we enforce

$$h(\beta_m^k) \geq 1, \quad m = 1, \dots, M_k,$$

which gives

$$\mathcal{K}^{(k)}(\beta_m^k)^\top \alpha^k \geq 1, \quad m = 1, \dots, M_k.$$

**Step 7: SOCP Reformulation.** Introduce an auxiliary variable $t_k$ such that

$$\sqrt{\alpha^{k\top} \mathcal{K}^{(k)} \alpha^k} \leq t_k. \implies \|R^k \alpha^k\| \leq t_k$$

Where $(R^k)^\top R^k = \mathcal{K}^{(k)}$.

The final SOCP formulation becomes

$$\min_{x \in \mathcal{X}, \{\alpha^k, t_k\}_{k=1}^K} d(x, x_0)$$

subject to, for all $k \in [K]$,

$$\frac{1}{N_k} \mathbf{1}^\top \mathcal{K}_{N_k, J}^{(k)} \alpha^k + \epsilon_k t_k \leq \delta,$$

$$\|R^k \alpha^k\| \leq t_k,$$

$$\mathcal{K}^{(k)}(\beta_m^k)^\top \alpha^k \geq 1, \quad m = 1, \dots, M_k.$$

This formulation is convex and can be solved efficiently using standard SOCP solvers.

## B.2 QCP vs SOCP: Empirical Comparison

To assess the practical implications of the proposed SOCP reformulation, we compare it with the original QCP formulation of KDRRA across all datasets. Both formulations are evaluated using the same experimental setup. This analysis highlights the trade-off between solution tightness and computational tractability.

Table 8: Validity $(M_1, M_2)$ and cost $(l_1, l_2)$ for KDRRA variants on real datasets.

| Dataset | Method | $M_1$ | $M_2$ | $l_1$ | $l_2$ |
|---|---|---|---|---|---|
| German | KDRRA-QCP | $1.00 \pm 0.00$ | $1.00 \pm 0.00$ | $1.28 \pm 0.80$ | $0.64 \pm 0.40$ |
| | KDRRA-SOCP | $1.00 \pm 0.00$ | $1.00 \pm 0.00$ | $1.45 \pm 0.90$ | $0.72 \pm 0.45$ |
| SBA | KDRRA-QCP | $1.00 \pm 0.00$ | $1.00 \pm 0.00$ | $2.41 \pm 0.56$ | $1.26 \pm 0.24$ |
| | KDRRA-SOCP | $1.00 \pm 0.00$ | $1.00 \pm 0.00$ | $2.95 \pm 0.65$ | $1.40 \pm 0.30$ |
| Student | KDRRA-QCP | $1.00 \pm 0.00$ | $1.00 \pm 0.00$ | $1.98 \pm 0.21$ | $0.98 \pm 0.11$ |
| | KDRRA-SOCP | $1.00 \pm 0.00$ | $1.00 \pm 0.00$ | $2.20 \pm 0.25$ | $1.05 \pm 0.15$ |
| Adult | KDRRA-QCP | $1.00 \pm 0.00$ | $1.00 \pm 0.00$ | $3.56 \pm 0.89$ | $1.97 \pm 0.21$ |
| | KDRRA-SOCP | $1.00 \pm 0.00$ | $1.00 \pm 0.00$ | $4.10 \pm 0.91$ | $2.20 \pm 0.30$ |

# C Appendix: Proofs

## C.1 Proof of Theorem

4.8

*Proof.* Define $X_i := \phi(\theta_i) \in \mathcal{H}$, so that $\mathbb{E}[X_i] = \mu_{\star,k}$ and

$$\hat{\mu}_{n_k} - \mu_{\star,k} = \frac{1}{n_k} \sum_{i=1}^{n_k} (X_i - \mu_{\star,k}).$$

**Step 1: Expected deviation.** By Jensen's inequality,

$$\mathbb{E}\|\hat{\mu}_{n_k} - \mu_{\star,k}\|_{\mathcal{H}} \le \sqrt{\mathbb{E}\|\hat{\mu}_{n_k} - \mu_{\star,k}\|_{\mathcal{H}}^2}.$$

Expanding the squared norm and using independence,

$$\mathbb{E}\|\hat{\mu}_{n_k} - \mu_{\star,k}\|_{\mathcal{H}}^2 = \mathbb{E}\left\|\frac{1}{n_k}\sum_{i=1}^{n_k}(X_i - \mu_{\star,k})\right\|_{\mathcal{H}}^2$$
$$= \frac{1}{n_k^2}\sum_{i=1}^{n_k}\mathbb{E}\|X_i - \mu_{\star,k}\|_{\mathcal{H}}^2,$$

since the cross terms vanish due to independence and $\mathbb{E}[X_i - \mu_{\star,k}] = 0$. Hence,

$$\mathbb{E}\|\hat{\mu}_{n_k} - \mu_{\star,k}\|_{\mathcal{H}} \le \sqrt{\frac{1}{n_k}\mathbb{E}\|X_1 - \mu_{\star,k}\|_{\mathcal{H}}^2}.$$

Using $\mathbb{E}\|X_1 - \mu_{\star,k}\|^2 = \mathbb{E}\|X_1\|^2 - \|\mu_{\star,k}\|^2 \le \mathbb{E}\|X_1\|^2$ and Assumption 4.7,

$$\mathbb{E}\|\hat{\mu}_{n_k} - \mu_{\star,k}\|_{\mathcal{H}} \le \frac{\kappa_0}{\sqrt{n_k}}.$$

**Step 2: Concentration.** Let $f(\theta_1, \ldots, \theta_{n_k}) := \|\hat{\mu}_{n_k} - \mu_{\star,k}\|_{\mathcal{H}}$. Replacing $\theta_j$ by $\theta_j'$ yields

$$|f(\cdot) - f(\cdot')| \le \left\|\frac{1}{n_k}(\phi(\theta_j) - \phi(\theta_j'))\right\|_{\mathcal{H}} \le \frac{2\kappa_0}{n_k},$$

so $f$ satisfies bounded differences with $c_j = 2\kappa_0/n_k$. By McDiarmid's inequality McDiarmid et al. (1989) ,

$$\mathbb{P}(f - \mathbb{E}f \ge t) \le \exp\left(-\frac{2t^2}{\sum_j c_j^2}\right) = \exp\left(-\frac{n_k t^2}{2\kappa_0^2}\right).$$

**Step 3: Conclusion.** Setting $t = \kappa_0\sqrt{2\log(1/\delta)/n_k}$ yields, with probability at least $1 - \delta$,

$$\|\hat{\mu}_{n_k} - \mu_{\star,k}\|_{\mathcal{H}} \le \frac{\kappa_0}{\sqrt{n_k}} + \kappa_0\sqrt{\frac{2\log(1/\delta)}{n_k}} = \varepsilon_{n_k}(\delta).$$

The equivalence with $P_{\star,k} \in \mathcal{K}_{\varepsilon_{n_k}(\delta)}(\hat{P}_{n_k})$ follows directly from the definition of the MMD ball. $\qquad\square$

## C.2 Proof of Lemma 4.1

*Proof.* Define

$$\mu := \sum_{i=1}^{M} c_i\,\phi(\alpha_i), \qquad \mu_P := \frac{1}{N}\sum_{j=1}^{N}\phi(\beta_j).$$

The RKHS constraint in Lemma 4.1 is

$$\|\mu - \mu_{\mathbb{P}}\|_{\mathcal{H}}^2 \le \epsilon^2.$$

**Step 1: Norm expansion.** By bilinearity of the inner product,

$$\|\mu - \mu_{\mathbb{P}}\|_{\mathcal{H}}^2 = \langle\mu, \mu\rangle_{\mathcal{H}} - 2\langle\mu, \mu_{\mathbb{P}}\rangle_{\mathcal{H}} + \langle\mu_{\mathbb{P}}, \mu_{\mathbb{P}}\rangle_{\mathcal{H}}.$$

Using linearity,

$$\langle\mu, \mu_{\mathbb{P}}\rangle_{\mathcal{H}} = \frac{1}{N}\sum_{j=1}^{N}\langle\mu, \phi(\beta_j)\rangle_{\mathcal{H}}, \qquad \langle\mu_{\mathbb{P}}, \mu_{\mathbb{P}}\rangle_{\mathcal{H}} = \frac{1}{N^2}\sum_{j,\ell=1}^{N}\langle\phi(\beta_j), \phi(\beta_\ell)\rangle_{\mathcal{H}}.$$

Hence,

$$\|\mu - \mu_{\mathbb{P}}\|_{\mathcal{H}}^2 = \langle \mu, \mu \rangle_{\mathcal{H}} - \frac{2}{N} \sum_{j=1}^{N} \langle \mu, \phi(\beta_j) \rangle_{\mathcal{H}} + \frac{1}{N^2} \sum_{j,\ell=1}^{N} \langle \phi(\beta_j), \phi(\beta_\ell) \rangle_{\mathcal{H}}. \tag{26}$$

**Step 2: Convert inner products to kernel evaluations.** By the reproducing property of the RKHS induced by $\kappa$ (equivalently, $\langle \phi(u), \phi(v) \rangle_{\mathcal{H}} = \kappa(u,v)$), we can rewrite each term.

*(i) The first term.* We have

$$\langle \mu, \mu \rangle_{\mathcal{H}} = \left\langle \sum_{i=1}^{M} c_i \phi(\alpha_i), \sum_{i'=1}^{M} c_{i'} \phi(\alpha_{i'}) \right\rangle_{\mathcal{H}} = \sum_{i=1}^{M} \sum_{i'=1}^{M} c_i c_{i'} \langle \phi(\alpha_i), \phi(\alpha_{i'}) \rangle_{\mathcal{H}} = \sum_{i=1}^{M} \sum_{i'=1}^{M} c_i c_{i'} \kappa(\alpha_i, \alpha_{i'}).$$

Let $\mathcal{K}_\alpha \in \mathbb{R}^{M \times M}$ be the Gram matrix with entries $(\mathcal{K}_\alpha)_{ii'} := \kappa(\alpha_i, \alpha_{i'})$ and let $\mathbf{c} = [c_1, \ldots, c_M]^\top$. Then the above double sum is exactly

$$\langle \mu, \mu \rangle_{\mathcal{H}} = \mathbf{c}^\top \mathcal{K}_\alpha \mathbf{c}.$$

*(ii) The mixed term.* For each $j$,

$$\langle \mu, \phi(\beta_j) \rangle_{\mathcal{H}} = \left\langle \sum_{i=1}^{M} c_i \phi(\alpha_i), \ \phi(\beta_j) \right\rangle_{\mathcal{H}} = \sum_{i=1}^{M} c_i \langle \phi(\alpha_i), \phi(\beta_j) \rangle_{\mathcal{H}} = \sum_{i=1}^{M} c_i \kappa(\alpha_i, \beta_j).$$

Summing over $j = 1, \ldots, N$ yields

$$\sum_{j=1}^{N} \langle \mu, \phi(\beta_j) \rangle_{\mathcal{H}} = \sum_{j=1}^{N} \sum_{i=1}^{M} c_i \kappa(\alpha_i, \beta_j) = \sum_{i=1}^{M} c_i \left( \sum_{j=1}^{N} \kappa(\alpha_i, \beta_j) \right).$$

Let $\mathcal{K}_{\alpha\beta} \in \mathbb{R}^{M \times N}$ be the cross-Gram matrix with entries $(\mathcal{K}_{\alpha\beta})_{ij} := \kappa(\alpha_i, \beta_j)$ and let $\mathbb{1} \in \mathbb{R}^N$ denote the all-ones vector. Then the vector $\mathcal{K}_{\alpha\beta} \mathbb{1} \in \mathbb{R}^M$ has $i$-th entry $\sum_{j=1}^{N} \kappa(\alpha_i, \beta_j)$, and hence

$$\sum_{j=1}^{N} \langle \mu, \phi(\beta_j) \rangle_{\mathcal{H}} = \mathbf{c}^\top \mathcal{K}_{\alpha\beta} \mathbb{1}.$$

*(iii) The last term.* Similarly,

$$\sum_{j,\ell=1}^{N} \langle \phi(\beta_j), \phi(\beta_\ell) \rangle_{\mathcal{H}} = \sum_{j,\ell=1}^{N} \kappa(\beta_j, \beta_\ell) = \mathbb{1}^\top \mathcal{K}_\beta \mathbb{1},$$

where $\mathcal{K}_\beta \in \mathbb{R}^{N \times N}$ is the Gram matrix with entries $(\mathcal{K}_\beta)_{j\ell} := \kappa(\beta_j, \beta_\ell)$.

**Step 3: Assemble the quadratic form.** Substituting (i)-(iii) into 26 gives

$$\|\mu - \mu_{\mathbb{P}}\|_{\mathcal{H}}^2 = \mathbf{c}^\top \mathcal{K}_\alpha \mathbf{c} - \frac{2}{N} \mathbf{c}^\top \mathcal{K}_{\alpha\beta} \mathbb{1} + \frac{1}{N^2} \mathbb{1}^\top \mathcal{K}_\beta \mathbb{1}.$$

Thus, $\|\mu - \mu_{\mathbb{P}}\|_{\mathcal{H}} < \epsilon$ implies

$$\mathbf{c}^\top \mathcal{K}_\alpha \mathbf{c} - \frac{2}{N} \mathbf{c}^\top \mathcal{K}_{\alpha\beta} \mathbb{1} + \frac{1}{N^2} \mathbb{1}^\top \mathcal{K}_\beta \mathbb{1} \leq \epsilon^2,$$

which is exactly 7. This completes the proof. $\qquad\square$

### C.3 Proof of Proposition 4.2

*Proof.* Fix an action $x$ and a shift index $k$. We show that the worst-case violation problem 6 over the kernel ambiguity set admits the finite-dimensional formulation 8.

**Step 1: Discrete approximation of the adversarial distribution.** Following the empirical approximation approach of Zhu et al. (2021), we restrict the worst-case probability distribution over $\tilde{\theta}_k$ to be supported on a finite set

$$\{\alpha_i\}_{i=1}^M = \{\beta_1, \ldots, \beta_N, \gamma_1, \ldots, \gamma_Z\}, \qquad M = N + Z,$$

where $\{\beta_i\}_{i=1}^N$ are the observed parameter samples and $\{\gamma_j\}_{j=1}^Z$ are additional synthetic support points introduced to enrich the representation near critical regions. The distribution is parameterized by weights $c_i \geq 0$ satisfying $\sum_{i=1}^M c_i = 1$, and we denote $\mathbf{c} = [c_1, \ldots, c_M]^\top$.

This discrete approximation reduces the infinite-dimensional optimization over probability measures to a finite-dimensional optimization over the simplex. Under this representation, the violation probability becomes

$$\mathbb{P}\big(C_{\tilde{\theta}_k}(x) \leq 0\big) = \sum_{i=1}^M c_i \, \mathbb{I}(C_{\alpha_i}(x) \leq 0),$$

which corresponds to a weighted empirical estimate of the misclassification event.

**Step 2: Kernel mean embedding constraint.** For the discrete distribution supported on $\{\alpha_i\}_{i=1}^M$, the kernel mean embedding is

$$\mu = \sum_{i=1}^M c_i \, \phi(\alpha_i),$$

while the reference embedding is approximated empirically as

$$\mu_P = \frac{1}{N} \sum_{i=1}^N \phi(\beta_i).$$

Thus, the ambiguity-set membership condition $\|\mu - \mu_P\|_{\mathcal{H}} \leq \epsilon$ is equivalent to

$$\left\| \sum_{i=1}^M c_i \phi(\alpha_i) - \frac{1}{N} \sum_{i=1}^N \phi(\beta_i) \right\|_{\mathcal{H}} \leq \epsilon.$$

**Step 3: Quadratic form via Gram matrices.** By Lemma 4.1, the above RKHS norm constraint is equivalent to

$$\mathbf{c}^\top \mathcal{K}_\alpha \mathbf{c} - \frac{2}{N} \mathbf{c}^\top \mathcal{K}_{\alpha\beta} \mathbb{1} + \frac{1}{N^2} \mathbb{1}^\top \mathcal{K}_\beta \mathbb{1} \leq \epsilon^2.$$

**Step 4: Compact objective representation.** Define the indicator vector

$$\mathcal{I}(x) = \big[\mathbb{I}(C_{\alpha_1}(x) \leq 0), \ldots, \mathbb{I}(C_{\alpha_M}(x) \leq 0)\big]^\top.$$

Then the objective can be written as $\mathcal{I}(x)^\top \mathbf{c}$.

Combining the objective, the quadratic constraint, and the simplex constraints on $C$ yields the finite-dimensional problem 8. $\qquad\square$

### C.4 Proof of Lemma 4.3

*Proof.* Recall from Lemma 4.1 that the RKHS constraint is equivalently

$$\mathbf{c}^\top \mathcal{K}_\alpha \mathbf{c} - \frac{2}{N} \mathbf{c}^\top \mathcal{K}_{\alpha\beta} \mathbb{1} + \frac{1}{N^2} \mathbb{1}^\top \mathcal{K}_\beta \mathbb{1} \le \epsilon^2. \tag{27}$$

Rearranging terms gives

$$\mathbf{c}^\top \mathcal{K}_\alpha \mathbf{c} - \frac{2}{N} \mathbf{c}^\top \mathcal{K}_{\alpha\beta} \mathbb{1} \le \epsilon^2 - \frac{1}{N^2} \mathbb{1}^\top \mathcal{K}_\beta \mathbb{1}. \tag{28}$$

Since $\mathcal{K}_\alpha \succeq 0$ is a Gram matrix, it admits a Cholesky factorization $\mathcal{K}_\alpha = R^\top R$. Hence

$$\mathbf{c}^\top \mathcal{K}_\alpha \mathbf{c} = \mathbf{c}^\top R^\top R \mathbf{c} = \|R\mathbf{c}\|_2^2.$$

Define the scalars

$$s := \frac{1}{N} \mathbb{1}^\top \mathcal{K}_{\alpha\beta} \mathbf{c}, \qquad s' := \frac{1}{N^2} \mathbb{1}^\top \mathcal{K}_\beta \mathbb{1}.$$

Then 28 becomes

$$\|R\mathbf{c}\|_2^2 \le 2s + (\epsilon^2 - s'). \tag{29}$$

**Step 1: A standard SOC identity.** For any vector $u \in \mathbb{R}^m$ and any scalars $p, q \in \mathbb{R}$, the Lorentz cone $\mathcal{Q}_{m+2} = \{(y, t_1, t_2) : \|y\|_2 \le t_2, \ t_2 \ge 0\}$ admits the following equivalent representation:

$$\|u\|_2^2 \le p^2 - q^2 \quad \Longleftrightarrow \quad \begin{bmatrix} u \\ p \\ q \end{bmatrix} \in \mathcal{Q}_{m+2}. \tag{30}$$

Indeed, $\begin{bmatrix} u^\top & p \end{bmatrix}^\top \in \mathcal{Q}_{m+1}$ with parameter $|q|$ is equivalent to $\|u\|_2^2 + p^2 \le q^2$, and rearranging yields 30. This is the standard "difference of squares" SOC encoding (see Boyd & Vandenberghe (2004)).

**Step 2: Match 29 to the SOC identity.** Let

$$p := s + \frac{1}{2}(\epsilon^2 - s'), \qquad q := -\frac{1}{2}.$$

Then

$$p^2 - q^2 = \left( s + \frac{1}{2}(\epsilon^2 - s') \right)^2 - \left( \frac{1}{2} \right)^2. \tag{31}$$

Expanding the right-hand side gives

$$p^2 - q^2 = s^2 + s(\epsilon^2 - s') + \frac{1}{4}(\epsilon^2 - s')^2 - \frac{1}{4}.$$

Now observe that the SOC inclusion stated in Lemma **??** is not written in the $(u, p, q)$ form directly, but rather in an equivalent *shifted* cone form:

$$\begin{bmatrix} R\mathbf{c} \\ s \\ s \end{bmatrix} - \begin{bmatrix} \mathbb{0} \\ \frac{1}{2}s' - \frac{1}{2}\epsilon^2 + \frac{1}{2} \\ \frac{1}{2}s' - \frac{1}{2}\epsilon^2 - \frac{1}{2} \end{bmatrix} \in \mathcal{Q}_{M+2}.$$

Denote the two scalar components after shifting by

$$t_1 := s - \left( \frac{1}{2}s' - \frac{1}{2}\epsilon^2 + \frac{1}{2} \right) = s + \frac{1}{2}(\epsilon^2 - s') - \frac{1}{2},$$

$$t_2 := s - \left( \frac{1}{2}s' - \frac{1}{2}\epsilon^2 - \frac{1}{2} \right) = s + \frac{1}{2}(\epsilon^2 - s') + \frac{1}{2}.$$

Therefore the cone inclusion is equivalent to

$$
\begin{bmatrix} \mathbf{c} \\ t_1 \\ t_2 \end{bmatrix} \in \mathcal{Q}_{M+2} \quad \Longleftrightarrow \quad \|R\mathbf{c}\|_2^2 \leq t_2^2 - t_1^2, \tag{32}
$$

by the SOC identity 30.

**Step 3: Simplify the right-hand side.** Using $t_2 - t_1 = 1$ and $t_2 + t_1 = 2\left(s + \frac{1}{2}(\epsilon^2 - s')\right)$, we get

$$
t_2^2 - t_1^2 = (t_2 - t_1)(t_2 + t_1) = 1 \cdot \left(2s + (\epsilon^2 - s')\right) = 2s + (\epsilon^2 - s').
$$

Substituting into 32 yields

$$
\|R\mathbf{c}\|_2^2 \leq 2s + (\epsilon^2 - s'),
$$

which is exactly 29, hence equivalent to the original quadratic constraint 27. This proves that the quadratic constraint can be written as the stated Lorentz cone inclusion. $\qquad\square$

### C.5 Proof of Proposition 4.4

*Proof.* We derive the dual of the SOCP 4.3 using standard conic duality arguments.

**Step 1: Standard conic form.** Let $y := \mathbf{c} \in \mathbb{R}^M$. For a fixed action $x$, the vector $\mathcal{I}(x) \in \mathbb{R}^M$ is constant. Problem 4.3 is a maximization of a linear function subject to: (i) one Lorentz cone constraint in $\mathcal{Q}_{M+2}$, (ii) one affine equality $\mathbb{1}^\top y = 1$, (iii) nonnegativity constraints $y \geq 0$.

**Step 2: Dual variables.** Introduce dual variables

$$
(w, a_1, a_2) \in \mathcal{Q}_{M+2} \quad \text{for the cone constraint,}
$$
$$
\lambda \in \mathbb{R} \quad \text{for the equality constraint } \mathbb{1}^\top y = 1,
$$
$$
\rho \in \mathbb{R}_+^M \quad \text{for the nonnegativity constraint } y \geq 0.
$$

Since the Lorentz cone is self-dual, the dual variable associated with the cone constraint lies in the same cone.

**Step 3: Stationarity condition.** Forming the Lagrangian and collecting all terms involving $y = \mathbf{c}$, the vanishing of the gradient with respect to $y$ yields the stationarity condition

$$
-\mathcal{I}(x) = R^\top w + \frac{1}{N} \mathcal{K}_{\alpha\beta} \mathbb{1} \, a_1 + \frac{1}{N} \mathcal{K}_{\alpha\beta} \mathbb{1} \, a_2 + \mathbb{1}\lambda + I^\top \rho,
$$

which gives the equality constraint in 9.

**Step 4: Dual objective.** The dual objective is obtained by minimizing the Lagrangian over $y$ and collecting the constant terms arising from the shifted cone constraint and the affine equality. This yields

$$
-\left(\frac{1}{2N^2} \mathbb{1}^\top \mathcal{K}_\beta \mathbb{1} - \frac{\epsilon^2}{2} + \frac{1}{2}\right) a_1 - \left(\frac{1}{2N^2} \mathbb{1}^\top \mathcal{K}_\beta \mathbb{1} - \frac{\epsilon^2}{2} - \frac{1}{2}\right) a_2 - \lambda,
$$

which is exactly the objective in 9.

**Step 5: Dual cone constraint.** Membership of $(w, a_1, a_2)$ in the Lorentz cone $\mathcal{Q}_{M+2}$ is equivalent to the quadratic inequality

$$
\|w\|_2^2 + a_1^2 \leq a_2^2,
$$

which gives the second constraint in 9.

**Step 6: Elimination of $\rho$.** Since $\rho \geq 0$, the equality constraint in 9 is feasible for some $\rho$ if and only if

$$
R^\top w + \frac{1}{N} \mathcal{K}_{\alpha\beta} \mathbb{1} \, a_1 + \frac{1}{N} \mathcal{K}_{\alpha\beta} \mathbb{1} \, a_2 + \mathbb{1}\lambda \leq -\mathcal{I}(x),
$$

which yields the reduced dual formulation 10. This completes the derivation. $\qquad\square$

### C.6 Proof of Corollary 4.6

*Proof.* We show that replacing the indicator vector $\mathcal{I}_k(x)$ by a convex upper-bounding surrogate $\widetilde{\mathcal{I}}_k(x)$ yields a convex restriction of the feasible set of Theorem 4.5.

**Step 1: Pointwise dominance implies a conservative restriction.** By assumption, the surrogate $\psi : \mathbb{R} \to \mathbb{R}_+$ satisfies

$$\mathbb{I}(t \leq 0) \leq \psi(t) \qquad \forall\, t \in \mathbb{R}.$$

Applying this componentwise to $t = C_{\alpha_i^{(k)}}(x)$ yields

$$\mathcal{I}_k(x) \leq \widetilde{\mathcal{I}}_k(x) \qquad \text{(componentwise for all } x\text{)}.$$

Define the affine left-hand side vector

$$L_k(a_{1k}, a_{2k}, w_k, \lambda_k) := \frac{1}{N_k} \mathcal{K}_{\alpha\beta}^{(k)} \mathbb{1}\, a_{1k} + \frac{1}{N_k} \mathcal{K}_{\alpha\beta}^{(k)} \mathbb{1}\, a_{2k} + \mathbb{1}\lambda_k + R_k^\top w_k.$$

In the MIQCP formulation (Theorem 4.5), the constraint is of the form

$$L_k(a_{1k}, a_{2k}, w_k, \lambda_k) \leq -\mathcal{I}_k(x) \quad \Longleftrightarrow \quad L_k(a_{1k}, a_{2k}, w_k, \lambda_k) + \mathcal{I}_k(x) \leq 0.$$

Since $\mathcal{I}_k(x) \leq \widetilde{\mathcal{I}}_k(x)$, we have for every $x$,

$$L_k + \widetilde{\mathcal{I}}_k(x) \leq 0 \;\Rightarrow\; L_k + \mathcal{I}_k(x) \leq 0,$$

componentwise. Hence enforcing the surrogate constraint

$$L_k(a_{1k}, a_{2k}, w_k, \lambda_k) + \widetilde{\mathcal{I}}_k(x) \leq 0$$

is a sufficient (conservative) condition for feasibility of the original constraint, proving that the surrogate problem defines a restriction of the original feasible set.

**Step 2: Convexity of the surrogate constraint.** Each $C_{\alpha_i^{(k)}}(x)$ is affine in $x$ and that $\mathcal{X}$ is convex. Since $\psi$ is convex, the composition $\psi(C_{\alpha_i^{(k)}}(x))$ is convex in $x$ for each $i$. Therefore $\widetilde{\mathcal{I}}_k(x)$ is componentwise convex in $x$. Because $L_k(a_{1k}, a_{2k}, w_k, \lambda_k)$ is affine in $(a_{1k}, a_{2k}, w_k, \lambda_k)$, the mapping

$$(x, a_{1k}, a_{2k}, w_k, \lambda_k) \mapsto L_k(a_{1k}, a_{2k}, w_k, \lambda_k) + \widetilde{\mathcal{I}}_k(x)$$

is componentwise convex. Hence the inequality

$$L_k(a_{1k}, a_{2k}, w_k, \lambda_k) + \widetilde{\mathcal{I}}_k(x) \leq 0$$

defines a convex feasible set.

**Step 3.** The remaining constraint $\|w_k\|_2^2 + a_{1k}^2 \leq a_{2k}^2$ is nonconvex. The objective $d(x, x_0)$ is convex. Therefore, the surrogate formulation is a quadratically constrained program (QCP). $\qquad\square$

### C.7 Proof of Proposition 5.1

*Proof.* Fix any $P \in \mathcal{B}_{\varepsilon_k}^k(P_n^{\mathrm{pred}})$ and any $h \in \mathcal{H}$ such that $f(\cdot) \leq h(\cdot)$ pointwise. Then

$$\mathbb{E}_P[f] \leq \mathbb{E}_P[h]. \tag{33}$$

Add and subtract $\mathbb{E}_{P_n^{\mathrm{pred}}}[h]$ to obtain

$$\mathbb{E}_P[h] = \mathbb{E}_{P_n^{\mathrm{pred}}}[h] + \big(\mathbb{E}_P[h] - \mathbb{E}_{P_n^{\mathrm{pred}}}[h]\big). \tag{34}$$

To control the difference term, note that for any $h \in \mathcal{H}$ with $h \neq 0$, the scaled function $\bar{h} := h/\|h\|_{\mathcal{H}}$ satisfies $\|\bar{h}\|_{\mathcal{H}} = 1$. By the definition of the kernel discrepancy ($=$ MMD induced by $\mathcal{H}$),

$$\mathbb{E}_P[\bar{h}] - \mathbb{E}_{P_n^{\mathrm{pred}}}[\bar{h}] \leq D_\kappa(P, P_n^{\mathrm{pred}}). \tag{35}$$

Multiplying 35 by $\|h\|_{\mathcal{H}}$ yields

$$\mathbb{E}_P[h] - \mathbb{E}_{P_n^{\mathrm{pred}}}[h] \leq D_\kappa(P, P_n^{\mathrm{pred}}) \|h\|_{\mathcal{H}}. \tag{36}$$

If $h = 0$, then 36 holds trivially. Since $P \in \mathcal{B}_{\varepsilon_k}^k(P_n^{\mathrm{pred}})$, we have $D_\kappa(P, P_n^{\mathrm{pred}}) \leq \varepsilon_k$, and therefore

$$\mathbb{E}_P[h] - \mathbb{E}_{P_n^{\mathrm{pred}}}[h] \leq \varepsilon_k \|h\|_{\mathcal{H}}. \tag{37}$$

Combining 33, 34, and 37 gives

$$\mathbb{E}_P[f] \leq \mathbb{E}_{P_n^{\mathrm{pred}}}[h] + \varepsilon_k \|h\|_{\mathcal{H}},$$

for every feasible pair $(P, h)$ with $P \in \mathcal{B}_{\varepsilon_k}^k(P_n^{\mathrm{pred}})$ and $h \in \mathcal{H}$ satisfying $f \leq h$. Taking the supremum over $P \in \mathcal{B}_{\varepsilon_k}^k(P_n^{\mathrm{pred}})$ on the left-hand side preserves the inequality, and then taking the infimum over all admissible $h$ on the right-hand side yields

$$\sup_{P \in \mathcal{B}_{\varepsilon_k}^k(P_n^{\mathrm{pred}})} \mathbb{E}_P[f] \leq \inf_{h \in \mathcal{H}, \, f \leq h} \left\{ \mathbb{E}_{P_n^{\mathrm{pred}}}[h] + \varepsilon_k \|h\|_{\mathcal{H}} \right\},$$

which is the desired claim. $\qquad\square$

### C.8 Proof of Lemma 5.3

*Proof.* Fix $(\kappa, x)$. Consider the optimization problem in Lemma 5.2, namely

$$\inf_{h \in \mathcal{H}, \, f_x \leq h} \left\{ \mathbb{E}_{P_n^{\mathrm{pred}}}[h] + \varepsilon_k \|h\|_{\mathcal{H}} \right\}. \tag{38}$$

To obtain a finite-dimensional characterization, we adopt the same finite set of kernel sites used in the empirical approximation (posterior sites and near-boundary sites) and enforce the domination constraint only at these locations. Concretely, let $\{\xi_j\}_{j=1}^J$ denote the chosen kernel centers and let $\{\zeta_m\}_{m=1}^M$ denote the sites at which we impose the pointwise constraints $f_x(\zeta_m) \leq h(\zeta_m)$. (In the main text, these are the boundary-adjacent sites, for which the constraint becomes $h(\zeta_m) \geq 1$.)

With this discretization, the feasible set is described by finitely many pointwise inequalities of the form

$$h(\zeta_m) \geq f_x(\zeta_m), \qquad m = 1, \dots, M, \tag{39}$$

and the objective contains the linear functional $h \mapsto \mathbb{E}_{\mathbb{P}_n^{\mathrm{pred}}}[h]$. In practice, $\mathbb{E}_{P_n^{\mathrm{pred}}}[h]$ is also approximated by a finite sample average over posterior-predictive draws $\{\tilde{\theta}_k^{(b)}\}_{b=1}^{B_k}$ i. e.,

$$\mathbb{E}_{P_n^{\mathrm{pred}}}[h] \approx \frac{1}{B_k} \sum_{b=1}^{B_k} h(\tilde{\theta}_k^{(b)}), \tag{40}$$

which is again a finite collection of evaluation functionals.

**Step 1: Decomposition into a data-dependent subspace and its orthogonal complement.** Let $\mathcal{S} \subset \mathcal{H}$ be the finite-dimensional subspace

$$\mathcal{S} := \mathrm{span}\{ \kappa(\cdot, \xi_1), \dots, \kappa(\cdot, \xi_J) \},$$

and decompose any $h \in \mathcal{H}$ as

$$h = h_{\mathcal{S}} + h_\perp, \qquad h_{\mathcal{S}} \in \mathcal{S}, \;\; h_\perp \in \mathcal{S}^\perp.$$

By the reproducing property, for any $\eta \in \mathcal{X}$,

$$h(\eta) = \langle h, \kappa(\cdot, \eta) \rangle_{\mathcal{H}}.$$

Since each evaluation point used in the objective and constraints belongs to the chosen set of kernel centers (or is included among them), $\kappa(\cdot, \eta) \in \mathcal{S}$, hence $h_{\perp}(\eta) = \langle h_{\perp}, \kappa(\cdot, \eta) \rangle_{\mathcal{H}} = 0$. Therefore,

$$h(\eta) = h_{\mathcal{S}}(\eta) \quad \text{for every evaluation point } \eta \text{ used in 39 and 40.} \tag{41}$$

In particular, replacing $h$ by $h_{\mathcal{S}}$ does not change the constraint values or the (approximated) expectation term in the objective.

**Step 2: Norm reduction.** Using orthogonality,

$$\|h\|_{\mathcal{H}}^2 = \|h_{\mathcal{S}}\|_{\mathcal{H}}^2 + \|h_{\perp}\|_{\mathcal{H}}^2 \geq \|h_{\mathcal{S}}\|_{\mathcal{H}}^2,$$

with strict inequality whenever $h_{\perp} \neq 0$. Since the objective in 38 penalizes $\|h\|_{\mathcal{H}}$ with a nonnegative weight $\varepsilon_k$, any feasible $h$ can be replaced by $h_{\mathcal{S}}$ to obtain a function that (i) satisfies the same discretized constraints and has the same linear evaluation terms by 41, but (ii) has a no larger RKHS norm. Hence there exists an optimizer lying entirely in $\mathcal{S}$.

**Step 3: Finite-dimensional expansion.** Any $h \in \mathcal{S}$ admits a representation

$$h(\tilde{\theta}) = \sum_{j=1}^{J} \alpha_j \, \kappa(\tilde{\theta}, \xi_j)$$

for some coefficients $\alpha_1, \ldots, \alpha_J \in \mathbb{R}$. Therefore, an optimizer of the discretized envelope problem admits the stated expansion. The representer-style argument is inspired by techniques in Schölkopf et al. (2001). □

