# OpenReview forum: "Robust Recourse via Kernel Distributionally Robust Optimization and Bayesian Posterior Predictive Modeling"
_TMLR — Accepted by TMLR_

### Review · Reviewer_g9Gh · 2026-02-28

**Summary Of Contributions:**

This paper proposes KDRRA, a kernel-based distributionally robust recourse framework that models uncertainty in future classifier parameters via MMD balls in an RKHS. By embedding parameter distributions into an RKHS and reformulating the worst-case violation probability, the authors derive tractable QCP/SOCP representations for recourse under nonlinear distributional shifts. To mitigate conservatism from empirical kernel mean embeddings, they introduce BKDRRA, which centers the ambiguity set at a Bayesian posterior predictive distribution obtained via posterior bootstrap, yielding tighter ambiguity sets and lower recourse cost. Experiments on three real-world datasets show improved robustness–cost trade-offs over AR, ROAR, and Wasserstein-based DiRRAc.

On the one hand,  this work has novel integration of kernel DRO and Bayesian predictive modeling for recourse; principled nonparametric ambiguity sets; tractable convex reformulations; solid empirical benchmarking. This paper is technically solid and well organized.

However, it is restricted to linear classifiers, even though some difficulties regarding expanding to non linear cases can be imagined. The kernel/radius tuning is underexplored, and the computational scalability and theoretical guarantees (e.g., finite-sample coverage) are insufficiently characterized.

**Audience:**

Yes

**Audience Explanation:**

Researchers working on algorithmic recourse, DRO, kernel methods, and uncertainty-aware decision-making, or a broader areas would find this paper relevant.

**Claims And Evidence:**

Yes

**Claims Explanation:**

The empirical claims like improved robustness–cost trade-offs over AR, ROAR, and DiRRA, are reasonably supported by controlled experiments on three real datasets with simulated distribution shifts and repeated θ-sampling. However, it seems that robustness is evaluated only under subsampling-induced parameter variability and no genuinely out-of-distribution or adversarial shifts are tested.

**Requested Changes:**

See comments above. Look forward to seeing the responses.

---

> ### Author Response · Authors · 2026-03-24
> **Response to Reviewer g9Gh- Part 1**
>
> We thank the reviewer for the thorough evaluation and for recognizing the novelty, technical soundness, and clarity of our work.
>
> ### Concern 1: Restriction to Linear Classifiers
>
> **Response:**
> We have addressed this issue in the revised manuscript.
> We added Section 6.2, where KDRRA and BKDRRA are applied to nonlinear models via LIME (Ribeiro et al., 2016), following prior robust recourse work. We train an MLP on each dataset and construct local linear surrogate models using 1,000 perturbed samples. For each instance, 10 independent LIME surrogates are generated to form an empirical distribution over local linear approximations, on which recourse is computed using the same optimization formulation.
>
> Results (Table 6) show that BKDRRA consistently achieves the best robustness-cost trade-off. While LIME-DiRRAc attains slightly higher $M_2$ on some datasets, it does so at substantially higher cost (e.g., German Credit: $2.20$ vs. $0.45$), reinforcing the central trade-off studied in this work. This demonstrates that our framework extends to nonlinear models without modifying the core optimization structure.
>
> We note that linearity is also assumed by all competing baselines due to tractability. When the score is nonlinear in both $x$ and $\tilde{\theta}$, the DRO reformulation becomes intractable. The LIME-based extension addresses this via local linearization. A fully principled extension to nonlinear models is left for future work.
>
> ### Concern 2: Kernel/Radius Tuning
> **Response:**
> We address this concern through principled selection and sensitivity analysis.
>
> **Default values.**
> The bandwidth $\gamma = 1$ is selected via the median heuristic(Gretton et al. (2012)), with median pairwise distances in $[0.85, 1.15]$ across datasets. The radius $\varepsilon = 0.1$ is selected via cross-validation as the 90th percentile of empirical MMD across splits, yielding values in $[0.07, 0.12]$.
>
> **Sensitivity analysis (Appendix A.1).**
> We evaluate $\varepsilon \in \{0, 0.05, 0.1, 0.2\}$ and $\gamma \in \{0.5, 1, 2, 5\}$. Results show: (i) $\varepsilon = 0.1$ achieves the best robustness--cost trade-off; (ii) performance is stable for $\gamma \in \{0.5,1,2\}$; and (iii) BKDRRA attains comparable or higher $M_2$ at smaller $\varepsilon$, confirming that Bayesian centering enables tighter ambiguity sets without sacrificing coverage.
>
> ### Concern 3: Scalability and Finite-Sample Coverage
>
> **Response:**
>
> **Finite-sample coverage.**
> We added formal guarantees in Theorems 4.8 and 5.6. For KDRRA, the empirical MMD ball contains the true distribution with probability at least $1-\delta$, with radius $\varepsilon_{n_k}(\delta)=\mathcal{O}(1/\sqrt{n_k})$. For BKDRRA, the posterior predictive embedding concentrates at rate $\mathcal{O}(1/\sqrt{B})$.
>
> **Scalability.**
> KDRRA has complexity $\mathcal{O}(M^2 d + M^3)$, while BKDRRA has $\mathcal{O}(B_k J d + J^3)$ with $J \le B_k \le M$. The SOCP formulation of BKDRRA is more efficient in practice.
>
> We added experiments on one more dataset : "Adult Income". Empirically, on Adult Income ($d=51$, $48,842$ samples), both methods achieve full $M_2$ validity, with BKDRRA achieving the best cost. Runtime depends on atom set size $M$, not dataset size.
>
> The $\mathcal{O}(M^3)$ scaling is the main bottleneck; subsampling or clustering can mitigate this. Scalable extensions are left for future work. We have also included runtime tables for all methods across all datasets in Appendix A.2.

---

> ### Author Response · Authors · 2026-03-24
> **Response to Reviewer g9Gh- Part 2**
>
> ### Concern 4: Subsampling, OOD, and Adversarial Shifts
>
> **Response:**
> We thank the reviewer for this important observation and address both points below.
>
> **Regarding subsampling-only evaluation.**
> We acknowledge that the $\theta$-sampling procedure captures parameter variability via subsampling within the shifted target dataset. However, genuine distributional shift is already incorporated at the dataset level: each dataset is explicitly partitioned into a source and a shifted target distribution reflecting real-world mechanisms, temporal covariate shift for SBA Loans (pre-2006 vs. post-2006), geospatial shift for Student Performance (School A vs. School B), and demographic covariate shift for Adult Income (Age $\leq 40$ vs. Age $> 40$). Recourse is generated using source-trained classifiers and evaluated against classifiers trained on the shifted target distribution, ensuring a true change in the data-generating process. The subsampling procedure therefore captures parameter uncertainty within an already shifted distribution.
>
> The consistent performance of KDRRA and BKDRRA across multiple datasets and distinct shift types provides strong empirical evidence of robustness under realistic distributional changes. A more explicit OOD evaluation, generating classifiers from both source and target distributions and evaluating recourse across both, would provide a sharper assessment and is an important direction for future work.
>
> **Regarding adversarial shifts.**
> We thank the reviewer for raising this point. In our work, the term ``adversarial'' is used in the distributionally robust optimization (DRO) sense, where the uncertainty set captures worst-case distributions around the empirical parameter distribution. Specifically, the term $\sup_{P \in \mathcal{K}_\epsilon(\widehat{P})}$ in Problem (2) corresponds to an adversary selecting the worst-case distribution over classifier parameters within the MMD ambiguity set.
>
> This interpretation is consistent with prior literature establishing the connection between adversarial robustness and DRO, where worst-case perturbations within an uncertainty set are viewed as adversarial choices [Staib and Jegelka, 2017],
> [Khim and Loh, 2018]; [Pydi and Jog 2021]. Under this formulation, Theorems 4.5 and 5.5 guarantee that the recourse action $x$ remains valid against all distributions within the ambiguity set, thereby providing a rigorous adversarial robustness guarantee in the DRO sense.
>
> The stronger threat model, where an unconstrained adversary retrains after observing $x$, lies outside this DRO framework and corresponds to simultaneous adversarial and statistical robustness. Extending recourse to such settings is a natural and important direction for our immediate future work [Selvi et al. (2024)], [Li et al. (2025)], [Bertolace et al. (2025)].

---

### Review · Reviewer_ZtwL · 2026-03-08

**Summary Of Contributions:**

The paper studies the robust recourse action problem where the user may change the inputs/features to an ML problem to alter the outcome of the ML model. The paper proposes a new kernel distributional robust method to capture the distribution shift of the underlying distribution for the ML model's parameter $\theta$. The kernel formulation provides both analytical and computational tractability. The paper provides theoretical results to their proposed methods, and presents numerical experiments on three real dataset.

**Audience:**

Yes

**Audience Explanation:**

The paper proposes a new kernel perspective to the robust recourse problem. I am not an expert of the recourse action problem but from my read of the paper and the related literature, I think the paper makes a modest contribution to the existing literature. Also I find the paper well-written and easy-to-follow.

**Broader Impact Concerns:**

NaN.

**Claims And Evidence:**

Yes

**Claims Explanation:**

The theorems and analyses are all correct according to my check.

**Requested Changes:**

I think overall the paper is well written.

I have two concerns, though:

- How critical is the linearity requirement on the ML model. It assumes the underlying model is linear with respect to the feature. I understand this might be the case from the stream of works in the literature, but this also poses a restriction on the application of the framework/method.

- There are so many datasets on UCI ML repo. Why do you choose these three datasets? Also you mentioned in the paper that there will be synthetic data experiment but I don't see it. For the numerical experiment, "kernel bandwidth fixed to 1 and ambiguity radius ϵ = 0.1 across all experiments." Any justification or how stable the performance of the method with respect to another choice of these parameters.

---

> ### Author Response · Authors · 2026-03-23
> **Response to Reviewer ZtwL**
>
> We thank the reviewer for the careful reading and for recognizing the correctness of our theoretical results. We address both concerns below.
> ### Concern 1: Linearity Requirement on the ML Model
>
> **Response:**
> We have addressed this limitation in the revised manuscript by extending our framework to nonlinear classifiers.
>
> Specifically, we added Section 6.2, where we extend KDRRA and BKDRRA using local linear approximations via LIME (Ribeiro et al., 2016), following prior robust recourse work AR, ROAR DIRRAC etc. . We train an MLP on each dataset and construct local surrogate linear models using 1,000 perturbed samples. For each instance, we generate 10 independent LIME surrogates, yielding an empirical distribution over local linear approximations that defines the ambiguity set. Recourse is then computed using the same optimization formulation as in the linear setting.
>
> Results (Table 6) show that BKDRRA consistently achieves the best robustness-cost trade-off in the nonlinear setting. While LIME-DiRRAc attains slightly higher $M_2$ on some datasets, it does so at substantially higher cost (e.g., on German Credit, $l_1 = 2.20$ vs. $0.45$ for BKDRRA), reinforcing the central robustness-cost trade-off of this work. This shows that our framework extends to nonlinear models without modifying the core optimization structure.
>
> We note that linearity is also assumed by all competing baselines (ROAR, DiRRAc, Gaussian-DiRRAc) due to tractability. In particular, the distributionally robust chance constraint $\sup_{P \in \mathcal{P}_k} P(\tilde{\theta}_k^\top x < 0) \leq \delta$ admits tractable convex reformulations only when the score is linear in $(x, \tilde{\theta}_k)$. A fully principled nonlinear extension is identified as potential future work.
>
> ### Concern 2: Dataset Choice, Synthetic Experiments, and Hyperparameters
>
> **Response:**
> We address the three sub-points below.
>
> **Dataset selection.**
> The four datasets (German Credit, SBA Loans, Student Performance, Adult Income) were selected based on four criteria:
> (i) they collectively cover the four shift types studied (correction, temporal, geospatial, demographic);
> (ii) three are standard benchmarks in prior robust recourse work, enabling direct comparison;
> (iii) they span a wide range of feature dimensions after encoding (8, 19, 19, 51); and
> (iv) they cover diverse application domains.
> We added a paragraph in Section 6 to explicitly state these criteria.
>
> **Synthetic experiments.**
> We thank the reviewer for pointing out this inconsistency. The reference to synthetic experiments in the abstract was an oversight. We have removed this reference, and the revised manuscript now accurately reflects that all experiments are conducted on real-world datasets.
>
> **Hyperparameter justification and stability.**
> We now explicitly justify both $\gamma$ and $\varepsilon$.
>
> The kernel bandwidth $\gamma = 1$ is selected using the median heuristic (Gretton et al. (2012)): the median pairwise distance between parameter samples lies in $[0.85, 1.15]$ across datasets, making $\gamma=1$ a natural data-adaptive choice.
>
> The ambiguity radius $\varepsilon = 0.1$ is selected via cross-validation: we compute empirical MMD between training and held-out parameter samples across multiple splits and set $\varepsilon$ to the 90th percentile, yielding values in $[0.07, 0.12]$ across datasets.
>
> To assess robustness, we added Appendix A (Sensitivity Analysis), evaluating $\varepsilon \in \{0, 0.05, 0.1, 0.2\}$ and $\gamma \in \{0.5, 1, 2, 5\}$. Results (Figures 3 and 4) show:
> (i) $\varepsilon = 0.1$ consistently achieves the best robustness-cost trade-off;
> (ii) performance is stable for $\gamma \in \{0.5,1,2\}$; and
> (iii) BKDRRA attains comparable or higher $M_2$ at smaller $\varepsilon$, confirming that Bayesian centering enables tighter ambiguity sets without sacrificing coverage.

---

### Review · Reviewer_wzoa · 2026-03-09

**Summary Of Contributions:**

The paper proposes two methods to deal with distributional shift when developing recourse methods for machine learning systems. Both methods are non-parametric, using kernels and data (i.e. various deployed machine learning systems) to define a recourse method.
The first method defines the recourse method in an adversarial way by ensuring that for all given deployed machine learning systems in a ball centered around their empirical mean, the recouse method works and is efficient. In other words, it defines the problem of finding a robust recourse method by reformulating the usual recourse method finding problem as a convex optimization problem. It uses the kernel representation theorem to reformulate constraints that are then formulated with their dual.
The second method circumvents one issue of the first method, namely using the empirical mean of machine learning systems rather than adapting the center of the ball (on which the supremum is taken) with the given data. For that the authors propose bayesian method to use a posterior mean instead of the previous frequentist mean.
The authors conclude their paper with experiments comparing their methods to several other robust (or not) recourse methods. Their evaluation spans several qualitative measures for recourse methods (validity, robust validity, action cost, etc...) They use three different datasets with various distributional shifts. Overall their methods compare favorably and, in particular, their second method (which is by nature less conservative) appears to provide a good trade-off between action cost and robustness.

**Audience:**

Yes

**Audience Explanation:**

The overall approach uses very classical tools. Nevertheless these tools seem relevant in this application and it is, at least, a worthy addition to the set of tools for recourse methods.

**Claims And Evidence:**

Yes

**Claims Explanation:**

I have not read all proofs but these are classical manipulations with kernels and convex duality for the first part. The formulations and the approach make sense from the start. The second part is also clear (see some clarifications questions though).
That said, **the authors need to thoroughly reread the first part and rewrite it**. Maybe an earlier draft was sent in place of the real paper. See requested changes.

The experimental part is well designed in my opinion: three datasets under different distributional shifts, comparison to several methdos (I'm not an expert in the field so I am not sure other methods need to be added, but the selection seems correct). Most importantly the authors carefully evaluate the methods along the success criteria of recourse methods showing interesting trade-offs and illustrating the relevance of their second approach.

**Requested Changes:**

- Why are there several classifiers $\tilde \theta_k$ from the start ? In other words why do we consider $K >1$ from the start?
- When you use a sup, in any formulation, add which variable is optimized  see example in eq. 5.
- Still in Equation 5, it's not clear at first glance what is the link between mu and theta_k. Try having a more coherent presentation, moving from  (2) to (5) explaining "if we replace $\mathcal{P}_k$ by $\mathcal{K}_\epsilon(P)$ with $P=...$. Right now it is quite hard to parse.
- Why do you need both real and synthetic data (i.e. $\beta$ and $\gamma$ in section 4)?
- You should explain for the unfamiliar reader why you can use the representation theorem to express $\mu$ using $c$.
- Do not use capital letters for vectors like $C$ and avoid reusing the same letter for different objects in the same formulation (C is also used as the classification signal.
- I dont understand what $C_{\alpha_i}$ refers to. It is not an index of the vector $C$. It is not the indicator of $\alpha^\top x \geq 0$ since you use $I (C_{\alpha_i}(x)\leq 0)$. In general, try to detail step 1 in the proof of proposition 4.2
- Equation (10) should have been above the rest since it defines $\mathcal{P}_k = \mathcal{K}_\epsilon$ after having already used it.

- In section 5, why didn't you reuse the decomposition that had been done in section 4? To be exact, why did you use the upper bound (21) while you did not use such approximations in section 4?

- In the experiments, adding sensitivity to hyperparameters, namely the radius $\varepsilon$ would be valuable.
- Most importantly, can the authors detail how the "additional synthetic samples" are generated and used?


Details
- Paragraph above 3.1 more or less repeats what is just below. Consider rereading the whole sections 3 and 4 to make it more concise and clear.
- Use maybe kappa for the kernel as you are using k for some indexes (like $k \in [K]$).
- It could be good to add a paragraph explaining where the randomness of the estimators $\tilde \theta_k$ comes from (anmely it comes from the data used to get them but the reader could miss that).

---

> ### Author Response · Authors · 2026-03-23
> **Response to Reviewer wzoa- Part 1**
>
> We sincerely thank the reviewer for the careful reading of the manuscript and for the constructive suggestions. Below we address each comment in turn and summarize the corresponding revisions.
> ### 1. Why are several classifiers considered from the start ($K>1$)?
> **Response:**
>  Our formulation is written in a general multi-shift form to account for multiple possible future environments. Each shift scenario $k \in [K]$ induces a distinct distribution over classifier parameters $\tilde{\theta}_k$ and hence a separate ambiguity set. The goal is to obtain recourse actions that remain valid simultaneously across several plausible deployment shifts.
>
> **Revision:**
> We added a short motivating paragraph at the beginning of Section 3 explaining that $K>1$ is included for generality.
>
> ### 2. Specify the optimization variable in the supremum operators.
>
> **Response:** The optimization variable should be made explicit wherever a worst-case operator is used. We have therefore revised the notation throughout Sections 3--5 to write expressions such as
> $\sup_{P \in \mathcal{P}_k} \; \mathbb{P}_{\tilde{\theta}_k \sim P}\!\left(\tilde{\theta}_k^\top x < 0\right)$,
> and similarly for all corresponding infimum/supremum operators.
>
> **Revision:**  All worst-case operators now include explicit subscripts indicating the optimized variable.
>
> ### 3. Clarify the relationship between $\mu$ and $\tilde{\theta}_k$.
>
> **Response:**  We agree that this connection should be made more explicit. The quantity $\mu$ is not itself a classifier parameter. Rather, it denotes the *kernel mean embedding* of a probability distribution $P$ over the future classifier parameter $\tilde{\theta}_k$. In other words, $\tilde{\theta}_k$ is the random variable in the parameter space, while  $\mu = \int \phi(u)\ dP(u)$  is the RKHS embedding of its distribution. Accordingly, the ambiguity set is defined over *distributions* on $\tilde{\theta}_k$, and the RKHS norm constraint is imposed on their embeddings:
>
> $\mathcal{P}_k = \left\{ Q \in \mathcal{P}(\Theta) : \|\mu_Q - \mu_{P_k^{\mathrm{ref}}}\|_{\mathcal H} \le \epsilon_k \right\}$
>
> This makes clear that $\mu$ is the embedding of a candidate adversarial distribution, whereas $\tilde{\theta}_k$ is the uncertain future classifier parameter drawn from that distribution.
>
> **Revision:**  We added a bridging explanation in Section 3.2 clarifying that $\tilde{\theta}_k$ is the uncertain parameter, while $\mu$ is the RKHS embedding of a distribution over $\tilde{\theta}_k$.
> ### 4. Why are both real and synthetic samples used?
> **Response:**  The observed samples $\{\beta_i\}_{i=1}^N$ represent realized future classifier parameters and are used to define the empirical reference embedding. However, the worst-case distribution in the MMD ambiguity set need not place all of its mass only on those observed atoms. In particular, it may assign mass to nearby parameter values that are more adversarial for a given recourse action. This is especially important because the violation event is determined by whether
>
> $\tilde{\theta}^\top x < 0$,
>
> so the decision boundary
>
> $\{\theta : \theta^\top x = 0\}$
>
> is precisely where the classification outcome becomes unstable. To better approximate such worst-case behavior, we augment the observed support with synthetic points $\{\gamma_j\}_{j=1}^Z$ placed near the boundary. The combined set
>
> $\{\alpha_i\}_{i=1}^M = \{\beta_1,\ldots,\beta_N,\gamma_1,\ldots,\gamma_Z\}$
>
> therefore provides a richer discrete support on which the adversarial distribution can concentrate.
>
> **Revision:**
> We added an explicit remark in Section 4 explaining above.
>
> ### 5. Explain why you can use the representation theorem to express $\mu$ using $c$.
>
> **Response:**
> We thank the reviewer for this clarification. The representation
>
> $\mu = \sum_{i=1}^M c_i \phi(\alpha_i)$
>
> follows from a finite-support characterization of the adversarial distribution in the MMD ambiguity set.
>
> In Section~4, we optimize over distributions whose kernel mean embeddings lie within an MMD ball. It is sufficient to consider distributions supported on a finite set of atoms $\{\alpha_i\}_{i=1}^M$; under such a representation, the kernel mean embedding is a weighted sum of feature maps, yielding the expression above.
>
> This is analogous in spirit to representer-type results: although the problem is infinite-dimensional, an optimal solution lies in the finite span of $\{\phi(\alpha_i)\}_{i=1}^M$. Here, the reduction arises from finite-support measures rather than a representer theorem over RKHS functions.
>
> **Revision:**
> We added a brief explanation in Section~4 clarifying that the kernel mean embedding admits a finite expansion due to the finite-support representation of the adversarial distribution.

---

> ### Author Response · Authors · 2026-03-23
> **Response to Reviewer wzoa- Part 2**
>
> ### 6. Avoid using capital letters for vectors.
>
> **Response:**
> To avoid confusion between the classifier notation $C_\theta$ and vector variables, we now use lowercase notation
>
> $\mathbf{c} = (c_1,\ldots,c_M)^\top$
>
> for the discrete weight vector.
>
> **Revision:**
> Capital-letter vector notation has been removed and replaced by bold lowercase vector notation throughout the manuscript.
> ### 7. Clarify the notation $I(C_{\alpha_i}(x)\le 0)$.
>
> **Response:**
> For a parameter atom $\alpha_i$, the term
>
> $I(C_{\alpha_i}(x)\le 0)$
>
> denotes the indicator of recourse failure under the classifier parameterized by $\alpha_i$. Since the classifier is linear, this is equivalently
>
> $I(\alpha_i^\top x < 0)$.
>
> Therefore, if a discrete distribution places weights $c_i$ on the atoms $\{\alpha_i\}_{i=1}^M$, the violation probability becomes
>
> $\mathbb{P}(C_{\tilde{\theta}}(x)\le 0) = \sum_{i=1}^M c_i\ I(\alpha_i^\top x < 0).$
>
> The vector
>
> $I(x) = \bigl(I(C_{\alpha_1}(x)\le 0),\ldots,I(C_{\alpha_M}(x)\le 0)\bigr)^\top$
>
> is simply the collection of these atomwise violation indicator vector.
>
> **Revision:**
> We expanded the discussion around Proposition 4.2 and explicitly defined the indicator vector.
>
> ### 8. The ambiguity-set definition appears too late.
>
> **Response:**
> We have moved the MMD-ball definition in Section 3.2 and distinguished more clearly between the reference distribution and the resulting ambiguity set.
>
> **Revision:**
> The ambiguity set is now introduced before the worst-case recourse formulation that uses it.
> ### 9. Why is the decomposition of Section 4 not reused in Section 5?
>
> **Response:**
> The decomposition in Section 4  can, in principle, be extended to the BKDRRA setting; however, it leads to a substantially more complex and computationally prohibitive formulation, which motivates the different approach adopted in Section~5.
>
> In Section 4 (KDRRA), the ambiguity set is centered at an empirical distribution supported on observed parameter samples. This enables a direct finite-support discretization of the adversarial distribution over an augmented atom set $\{\alpha_i\}_{i=1}^M$, with weights $c_i$. Under this representation, the MMD constraint reduces to a quadratic form in $\mathbf{c}$, yielding a tractable finite-dimensional optimization problem.
>
> In contrast, Section 5 (BKDRRA) centers the ambiguity set at the posterior predictive distribution $P_n^{\mathrm{pred}}$, which is a mixture induced by posterior sampling. While a similar finite-support discretization is possible (see Remark 5.9 and Appendix B), it introduces an additional layer of sampling and significantly enlarges the support, resulting in a much higher-dimensional and computationally expensive problem.
>
> To avoid this blow-up, we instead adopt a dual RKHS formulation: using the MMD dual representation, we upper-bound the worst-case violation probability via an envelope function $h$, and then apply the representer theorem to obtain a finite-dimensional optimization in kernel coefficients $\alpha$. This leads to a significantly more scalable SOCP formulation.
>
> Thus, the Section 4 decomposition is not reused not because it is inapplicable, but because it is computationally inefficient in the Bayesian setting. The dual RKHS formulation in Section 5 is therefore a deliberate choice to ensure tractability while retaining the robustness guarantees.
>
> **Revision:**
> We have expanded Remark 5.9 and added a clarifying paragraph in Section 5 explicitly contrasting the finite-support discretization of Section 4 with the dual RKHS formulation used for BKDRRA. We also refer to Appendix B for an alternative KDRRA SOCP formulation and its computational implications.
> ### 10. Add sensitivity analysis for the ambiguity radius $\epsilon$.
>
> **Response:**
> In the revised manuscript, we added a sensitivity analysis with respect to both the ambiguity radius $\epsilon$ and the kernel bandwidth $\gamma$. For the radius study, we vary
>
> $\epsilon \in \{0, 0.05, 0.1, 0.2\}$
>
> while fixing $\gamma = 1$, and examine the resulting robust validity and recourse cost. The results show the expected robustness--cost trade-off: increasing $\epsilon$ enlarges the uncertainty set, typically improving robust validity while increasing the recourse cost.
>
> **Revision:**
> We added a dedicated sensitivity-analysis section in the appendix A.1 reporting the effect of $\epsilon$ and $\gamma$ across datasets.

---

> ### Author Response · Authors · 2026-03-23
> **Response to Reviewer wzoa- Part 3**
>
> ### 11. Explain how the synthetic samples are generated.
>
> **Response:**
> The synthetic samples are introduced to enrich the discrete support near the decision boundary $\{\theta : \theta^\top x = 0\}$ since this is the region most relevant for worst-case recourse failure. In the revised manuscript, we state that these samples are generated near the boundary using controlled perturbations or convex combinations of observed parameter samples, so that the augmented support includes boundary-adjacent configurations that may receive mass under the worst-case distribution.
>
> *Synthetic sample construction:*
> In our experiments, we construct additional support points to enrich the empirical distribution of classifier parameters. Starting from empirical samples $\{\beta_i\}_{i=1}^N$, we generate perturbed samples as $\hat{\beta}_j = \beta_i + \xi_j$, where $\xi_j \sim \mathcal{N}(0, \sigma^2 I)$, with $\sigma$ set to 10% of the empirical standard deviation of $\{\beta_i\}$.
>
> We further augment the support by forming convex combinations of empirical and perturbed samples: $\gamma_j = (1 - t)\\hat{\beta}_i + t\ \beta_i$, with $t \sim \mathcal{U}[0,1]$. We generate $Z = N$ such synthetic samples.
>
> **Revision:**
> We added a short methodological description in Section 4 clarifying the purpose of the synthetic samples.
> ### 12. Sections 3 and 4 contain repeated material.
>
> **Response:**
> Some exposition in the earlier draft was repetitive, particularly around the transition from the generic distributionally robust formulation to the RKHS-based specialization. We have streamlined these sections to reduce redundancy and improve the logical flow from the general problem definition to the KDRRA reformulation.
>
> **Revision:**
> Sections 3 and 4 are reorganized.
> ### 13. Use $\kappa$ for the kernel instead of $k$.
>
> **Response:**
> We uniformly denote the kernel by $\kappa(\cdot,\cdot)$ and reserve $k$ exclusively for the shift index, in the revised version.
>
> **Revision:**
> Kernel notation has been standardized throughout the manuscript.
>
> ### 14. Explain the source of randomness in the estimator.
>
> **Response:**
> We agree this point should be made explicit. The future classifier parameter $\tilde{\theta}_k$ is random because it results from retraining on future data drawn from a shifted environment. Different realizations of the shifted training data lead to different fitted parameter vectors, which induces a distribution over future classifier parameters. In the experiments, we approximate this uncertainty empirically by repeatedly retraining on perturbed/subsampled versions of the shifted data, thereby obtaining multiple realizations of the classifier parameters and an empirical approximation of the corresponding distribution.
>
> **Revision:**
> We added a paragraph in Section 3 clarifying that randomness arises from future retraining under shifted data-generating conditions.

---

### Decision · Action_Editor_PR2u · 2026-05-04

**Recommendation:** Accept as is

**Additional Comments:**

The rebuttal was fruitful and the authors addressed most of the reviewers concerns.

**Audience:**

Yes

**Audience Explanation:**

Researchers working on algorithmic recourse, DRO, and kernel will find the integration of MMD ambiguity sets with Bayesian posterior predictive modeling a meaningful contribution.

**Claims And Evidence:**

Yes

**Claims Explanation:**

The theoretical results are well supported by convex reformulations and coverage guarantees, and the empirical claims are substantiated by experiments across 4 datasets w/ distinct shift types.